# Maternal aggression driven by the transient mobilisation of a dormant hormone-sensitive circuit

Stefanos Stagkourakis [1] ✉, Paul Williams[2], Giada Spigolon[1], Shreya Khanal[1], Katharina Ziegler [1], Laura Heikkinen[1,2], Gilberto Fisone [1] & Christian Broberger [1,2] ✉

Aggression is a sexually dimorphic behaviour. In some species, including the laboratory mouse, it is robustly expressed in males – while females are not aggressive in the non-puerperal state. However, during nursing, females exhibit maternal aggression, a dramatic yet transient shift in their social behaviour repertoire. This phenotypic change occurring in adulthood presents an opportunity to investigate whether sex-biased behavioural programs depend on mono- or di-morphic neural circuits. While maternal hormones are known to elicit nursing, their role in maternal aggression, particularly regarding target sites and cellular mechanisms, remains unclear. Here, we show that a molecularly defined subset of mouse ventral premammillary (PMv$^{DAT}$) neurons – with an established role in intermale aggression– transitions from quiescence to a hyperexcitable state during female lactation. The maternal hormones, prolactin and oxytocin, were found to excite these cells through pre- and post-synaptic electrophysiological actions. Gain- and loss-of-function experiments related to PMv$^{DAT}$ neuron activity bidirectionally influence maternal aggression, while PMv$^{DAT}$ neuron activation suppressed the expression of a competing social behaviour. This study identifies a sexually monomorphic neural substrate in mice capable of integrating hormonal cues, providing a likely mechanism that enables the transient access to a dormant behavioural program.

Aggression is a near-ubiquitous adaptive behaviour in animals and humans that serves multiple purposes and takes several forms, depending on the agent and context[1–8]. Intermale aggression is arguably the best-studied form of aggression found in nature, plays a role in establishing social hierarchy and access to resources and mating opportunities. In contrast, aggressive encounters initiated by female mammals are more rare, species dependent[9–12], and influenced by a number of factors including the animal's hormonal profile[13], neuromodulation[14], stress levels[15], social experience, and photoperiod[16]. Female interactions with conspecifics, however, change dramatically after pregnancy and parturition, when lactating dams typically exhibit profound aggressive behaviour towards both males and females of the same species[17]. Indeed, aggression is as indelible as other components of maternal behaviour, such as nursing and nest-building. This dramatic–yet reversible[18]–phenotypic metamorphosis powerfully exemplifies a neurobiological challenge, specifically: How does the brain reconfigure to transiently enable the expression of a behaviour that is otherwise outside of the animal's repertoire? The brain loci and mechanisms underlying maternal aggression, and the signals that drive this transition, are underexplored and remain

[1]SciLifeLab, Department of Neuroscience, Karolinska Institutet, Solna, Sweden. [2]Department of Biochemistry and Biophysics, Stockholm University, Stockholm, Sweden. ✉e-mail: stefanos.stagkourakis@scilifelab.se; christian.broberger@dbb.su.se

ambiguous. A role for hormones that drive the maternal state has been proposed[19–27], but the mechanisms at the cellular and network level have received scant attention. Importantly, it is not well understood if the neural modules that lead to the expression of aggression are shared between the two sexes or exhibit sexual dimorphism[12,14,15]. Finally, it is unknown if brain nodes that elicit certain maternal behaviours facilitate or impair the expression of other behaviours typical of the mother.

Here we address the above questions by focusing on the ventral premammillary nucleus (PMv), which is coextensive with the caudal two-thirds of the classically defined "hypothalamic attack area"[28], and which has been implicated in social behaviour[29–31] and intermale aggression[32–35]. This nucleus receives dense innervation from areas associated with sensory processing[36], and projects[37] to nuclei implicated in social behaviour[38–41], aggression[33,42–45], and reward[34,46–49]. Within the PMv, a subset of glutamatergic cells that express the dopamine transporter gene (PMv^DAT neurons) have been specifically implicated in male agonistic behaviours[32,34,35].

Here we identify PMv^DAT cells as quiescent in the virgin female mouse brain, with a dramatic switch in their excitability during motherhood. These neurons are shown to be sensitive to the hormones that surge during nursing and which facilitate the expression of maternal care. Specifically, prolactin (Prl) triggers excitatory pre- and post-synaptic changes, an effect mimicked by another maternal hormone, oxytocin (OT)[50]. Optogenetic manipulation and deletion of PMv^DAT neurons revealed that their activity is instrumental in the expression of maternal aggression. Finally, activation of PMv^DAT neurons during a maternal care paradigm impairs the expression of pup

retrieval and nursing, suggesting a specialised role of PMv cells in prioritising a distinct behavioural expression at the expense of others.

The present study suggests that the transient sex-specific expression of an adaptive behaviour can result from state-dependent reversal of neuronal excitability.

## Results

### Low *vs.* high baseline activity of PMv^DAT cells in virgin *vs.* post-partum females

While maternal aggression is widespread in the animal kingdom, recent studies suggest substantial inter-individual variability in the expression of aggressive behaviour[19,51]. Furthermore, a number of variables impact on the expression of maternal aggression, including litter size[52], pup presence[53], and the intruder's sex[54] and behaviour[55]. To control for these factors, we used a three consecutive days resident-intruder (RI) paradigm to screen lactating dams for aggressive behaviour, using C57 or BALB/c juvenile virgin female mice (P40-50) as intruders and restricting litter size to four to six pups, on postnatal day three to five (P3-5). Under this paradigm, 42% of multiparous lactating dams expressed maternal aggression (Fig. 1a), with increased duration and decreased latency upon repeated exposure (Fig. 1b). The maternal aggression behaviours are similar to those observed in intermale aggression using the same paradigm[34,49,56], and are in proportions consistent with earlier descriptions of maternal aggression[57]. C57 virgin female mice exposed to the same experimental design (Fig. S1a) did not exhibit aggression-related behaviours, including chasing, or biting: a similar observation to previous studies[9,58].

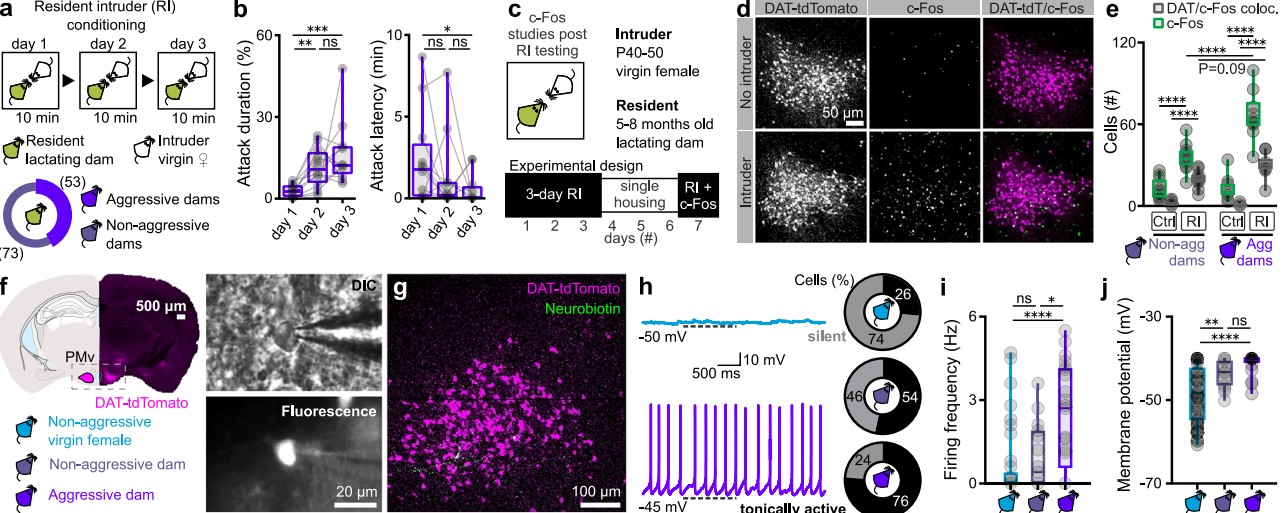

**Fig. 1 | PMv^DAT neurons activated by social encounters are electrically hyperexcitable in aggressive dams during the nursing state. a** Schematic representation of the experimental design used to identify lactating dams expressing maternal aggression (*n* = 126 lactating mice). Lactating dams were between five and eight months of age. **b** Quantification of maternal aggression traits identify increased attack duration (left; shown as the percentage of total session time) and decreased attack latency (right) with consecutive trials in the resident-intruder (RI) paradigm shown in Fig. 1A (*n* = 15 mice per group, RM one-way ANOVA with Tukey's test for correcting for multiple comparisons, ANOVA *P* value = 0.0010). **c** Schematic representation of the experimental design used to perform c-Fos studies following RI testing. **d** Confocal images of c-Fos immunoreactivity in the PMv post-RI testing in the absence (top panels) or presence (bottom panels) of a juvenile female conspecific. **e** Quantification of c-Fos and DAT-tdTomato positive cells in the PMv following control and social conditions, in lactating aggressive and non-aggressive mice (*n* = 12 sections from 6 mice without intruder exposure, and *n* = 10 sections from 5 mice with intruder exposure; comparisons performed using ordinary one-way ANOVA with Tukey's test for correcting for multiple comparisons, ANOVA *P*

value < 0.0001). **f** Identification of PMv^DAT neurons using the DAT-tdTomato mouse line to perform whole-cell patch clamp slice recordings from age-matched adult virgin female and lactating mice (five to eight months old). **g** PMv^DAT neurons were identified and reconstructed following the recording session using Neurobiotin-FITC. **h** Example recordings from PMv^DAT neurons (left) and group data in pie charts (right) from virgin female (blue) and lactating (purple) mice, exhibiting low (gray) *vs.* high (black) baseline spike discharge, respectively (*n* = 53 and 21 recorded cells from 26 virgin female and 11 lactating dams, respectively). **i** Quantification of firing frequency in PMv^DAT cells in virgin females *vs.* lactating dams (*n* = 53 *vs.* 21 cells, two-tailed unpaired t-test, *P* < 0.0001, *P* = 0.0104, and *P* = 0.7737 respectively). **j** Quantification of membrane potential in PMv^DAT cells in virgin females *vs.* lactating dams (*n* = 53 *vs.* 21 cells, two-tailed unpaired t-test, *P* < 0.0001, *P* = 0.0066, and *P* = 0.4106 respectively). See also Fig. S1. All box plots show the median (center line), 25th and 75th percentiles (box bounds), and minima/maxima (whiskers). Exact *P* values are provided where *P* > 0.0001; for *P* ≤ 0.0001, significance is indicated using asterisks (**** for *P* ≤ 0.0001, *** for *P* ≤ 0.001, ** for *P* ≤ 0.01, * for *P* ≤ 0.05, ns = not significant). Source data are provided as Source Data file.

PMv neurons have been reported to activate in conjunction with an aggressive encounter in lactating rat dams[31]. To determine if this includes the PMv[DAT] neurons, we examined c-Fos immunoreactivity. Both the total number of c-Fos[+] neurons, and the proportion of DAT[+] cells containing c-Fos immunoreactivity, were increased following the expression of maternal aggression (Fig. 1c–e). Increased c-Fos immunoreactivity was also observed in DAT[+] neurons in non-aggressive lactating dams (Fig. 1e) and in virgin female mice (in the estrus phase of the estrous cycle; Fig. S1a, b) following an encounter with an intruder. These observations are in agreement with reports in male rodents that PMv[DAT] neurons activate in variable degrees following different types of social encounters[32,34].

A subpopulation of male laboratory rodents are non-aggressive[34,49,56]. Strikingly, PMv[DAT] neurons in male non-aggressor mice are significantly less electrically excitable than in aggressors[34]. This observation prompted the question of whether phenotypic differences in female mice (virgin, lactating non-aggressive, and lactating aggressive) correlate with a distinct electrical profile of PMv[DAT] cells[59,60]. Brain slice whole-cell patch-clamp recordings were performed on PMv[DAT] neurons, which were *post-hoc* reconstructed (Fig. 1f, g). Notably, PMv[DAT] cells from lactating dams were in a hyperexcited state compared to those from virgin females and non-aggressive lactating dams (Fig. 1h–j), exhibiting a greater proportion of discharging *vs.* quiescent cells (Fig. 1h), significantly higher firing frequency (Fig. 1i) and a more depolarised resting membrane potential (Fig. 1j).

### Bidirectional control of maternal aggression through PMv[DAT] neurons

Having established that PMv[DAT] neurons are in a hyperexcitable state in lactating dams compared to the non-aggressive virgin female mouse, we next performed activation, inactivation and deletion of these cells in order to investigate the effect of such manipulations in vivo on maternal aggression. Genetic modifications were introduced into PMv[DAT] neurons by stereotactic injection of Cre-dependent constructs within an AAV vector.

PMv[DAT] neurons in lactating dams were transduced with channelrhodopsin-2 (ChR2, Fig. 2a) and behaviour was recorded in RI trials in the presence of male and female conspecifics (Fig. 2b). Minimal control-condition maternal aggression was attained in this experimental design by introducing an adult intruder (typically five months old and 30–35 grams in weight), in agreement with previous literature[61,62], and baseline behaviour was characterised by close investigation, but rare expressions of aggression (Fig. 2c). Optogenetic stimulation of PMv[DAT] cells resulted in significantly prolonged attack bouts appearing within seconds of initiation of photostimulation (Fig. 2c–e, Video S1). In marked contrast, photoactivation of PMv[DAT] neurons in virgin female mice (Fig. S1c–g) and in non-aggressive lactating dams (Fig. S1h, i), did not lead to aggression, but rather an increased amount of close investigation of a conspecific. The latter experiment suggests that the ability of an activated PMv[DAT] population to trigger aggression is state-dependent.

To determine the behavioural effect of inhibition of the PMv[DAT] population, these cells in lactating dams were transduced with halorhodopsin-3 (eNpHR3, Fig. 2f) and juvenile female intruders were used to facilitate high levels of maternal aggression in baseline conditions[54,62] (Fig. 2g). Indeed, under these control conditions, lactating dams frequently engaged in sustained attacks. However, optogenetic inhibition of their PMv[DAT] neurons was followed by decreased attack bout duration and delayed initiation of subsequent attack episodes (Fig. 2h–j).

Lastly, genetic deletion of the PMv[DAT] neurons in lactating dams (sparing neighbouring DAT neuron populations; Fig. S2a, b) and subsequent RI testing (Fig. 2k, l) dramatically decreased the attack frequency and duration and increased latency to attack (Fig. 2m–o), in agreement with an earlier report applying non-specific excitotoxic

PMv lesions[31]. In addition, genetic deletion of the PMv[DAT] neurons altered the dams' sociability in non-aggression related social encounters (Fig. S2c), but had no effect on exploration-related behaviours (Fig. S2d). Together, these experiments establish a causal and state-dependent role for PMv[DAT] neurons in the expression of maternal aggression.

### Maternal hormones excite PMv[DAT] neurons

Endocrine hallmarks of the postpartum female include a surge in the serum levels of specific hormones[63–67]. Among these hormones, the most well-documented maternal functions are served by Prl[68], which rises in late pregnancy and remains elevated after birth in rodents[69–71], facilitating maternal pup care behaviour and offspring survival[72–74]. Similar dynamics are shown by OT, which plays a parallel and complementary role in the mother's care for offspring[75–78]. We next explored if these maternal hormones can affect the activity of PMv[DAT] neurons that the data above implicate in puerperal aggression.

Serum levels of Prl were found ten-fold higher in lactating mouse dams compared to virgin females at any phase of the estrous cycle (Fig. 3a), confirming earlier literature[79]. We next determined if this difference in circulating Prl corresponds to differential Prl receptor (Prl-R) activation in the PMv of virgin *vs.* lactating female mice. Indeed, phosphorylation of the Signal Transducer and Activator of Transcription 5 (pSTAT5), a key sensitive downstream mediator of Prl-R activation[80–91], was significantly elevated in PMv[DAT] neurons in lactating females (Fig. 3b–d). These findings prompted the question of whether PMv[DAT] cells can be directly modulated by Prl.

The application of Prl to PMv[DAT] neurons during ex vivo patch clamp recordings yielded several electrophysiological changes, which can increase the excitability and output of these cells. Recordings were performed on brain slices collected from virgin female mice, in order to investigate the effect of the hormone on these cells without prior exposure to the long-term high Prl and OT concentrations associated with the puerperal state[70], which could lead to receptor desensitisation, a feature of peptide receptors[89,91]. Firstly, a prominent and reversible depolarisation, paralleled by increased discharge frequency, was seen (Fig. 3e), similar to observations from genetically unidentified PMv neurons[92]. Secondly, action potential duration was significantly prolonged (Fig. 3f), to levels that have been reported to increase neurotransmitter release by an order of magnitude[93]. Thirdly, the capacity for, and strength of, post-inhibitory rebound were augmented (Fig. 3g), key features of PMv[DAT] neurons, and implicated in their propensity for regenerative discharge[34]. The post-synaptic Prl-induced current in PMv[DAT] cells consisted of a low- and a high-voltage component (Fig. 3h). Application of the selective T-type low-threshold $Ca^{2+}$ current blocker, ML218, abolished the low-voltage component (Fig. 3i), and the post-inhibitory rebound (Fig. 3g). Lastly, we tested whether blocking the T-type $Ca^{2+}$ channels impaired the post-inhibitory rebound and the Prl-induced potentiation of rebound excitation. Indeed, ML218 abolished the post-inhibitory rebound current and effect of Prl at below firing threshold membrane potential (Fig. 3g).

Interestingly, the application of Prl also affected synaptic input into PMv[DAT] neurons, with an increase of both spontaneous excitatory and inhibitory post-synaptic currents (Fig. 3j). A likely source of the increased excitatory synaptic currents is recurrent axon collaterals, as the glutamatergic PMv[DAT] population is densely interconnected[34]. These data reveal a wide repertoire of Prl actions on the membrane properties of PMv[DAT] neurons that together converge to increase excitability.

Oxytocin, beyond its role in milk ejection[94–96], is most commonly discussed as an agent of maternal care and social bonding[97,98]. Previous studies have, however, also correlated increased serum OT levels to maternal aggression[19,26,99]. Yet, a cellular mechanism for how OT might promote attack behaviour in dams remains elusive. Immunofluorescence staining of the PMv identified close appositions between

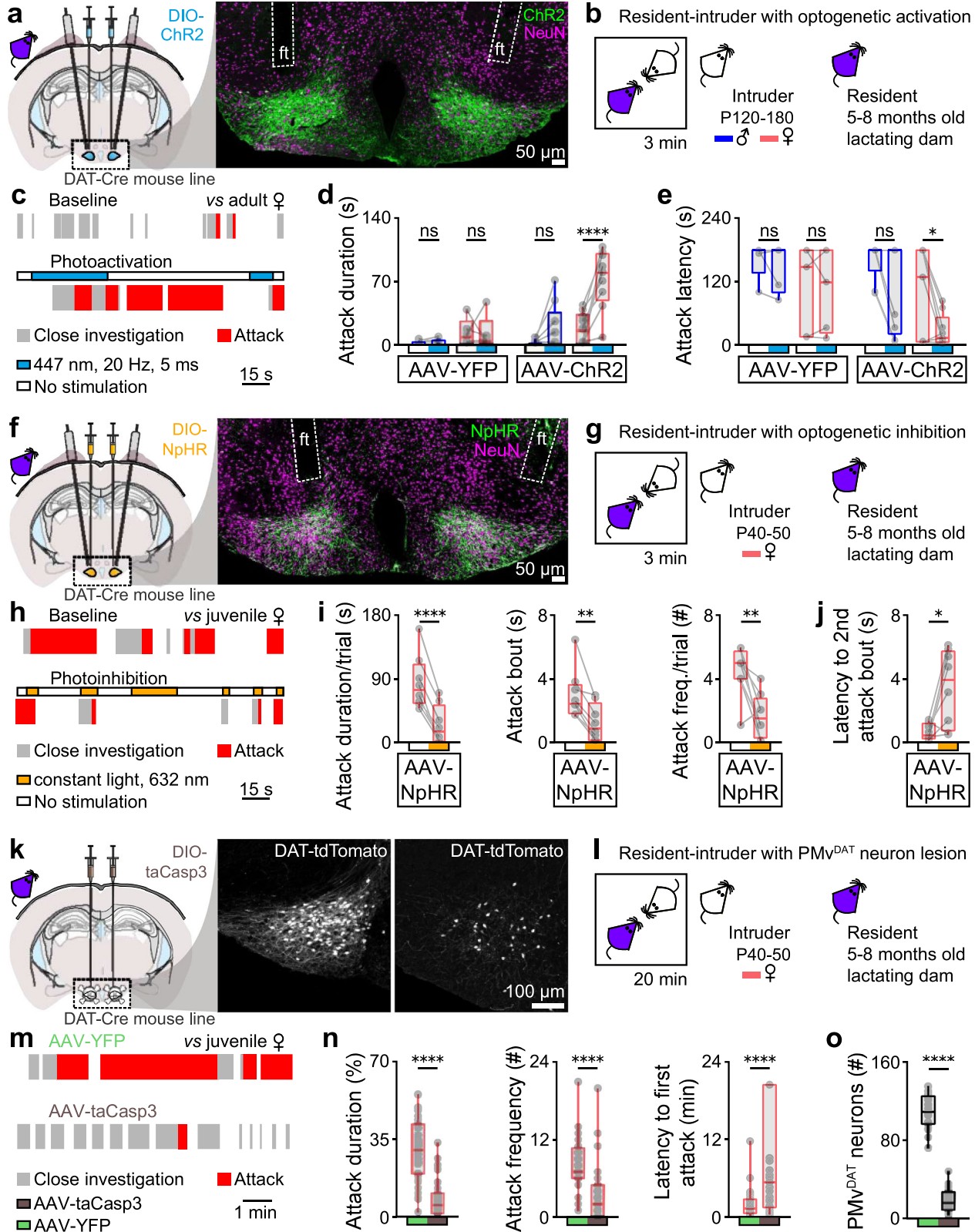

OT positive fibers and PMv[DAT] cell somata (Fig. 3k). This finding, and the presence of OT receptors (OT-R's) in the PMv[100], led us to evaluate the responsiveness of PMv[DAT] neurons to OT. Application of OT yielded a reversible depolarisation of PMv[DAT] neurons (Fig. 3l), similar to the effect of Prl (as above), as did application of the selective OT-R agonist, TGOT[101,102] (Fig. 3m). However, in the presence of an OT-R antagonist (vasotocin [d(CH2)$_5$[1], Tyr(Me)[2], Thr[4], Orn[8], des-Gly NH2[9]])[100] a

depolarisation of resting membrane potential was still observed when OT was applied. Thus, although available histochemical data from mice and rats indicate the expression of OT-R's, but not vasopressin receptors in the PMv[100,103,104], (in situ hybridisation experiment against the Avpr1a mRNA across the C57BL/6J mouse brain), these pharmacological results suggest that vasopressin receptors[105–111] may also contribute to the peptide's action on PMv[DAT] neurons (Fig. 3m). The OT

**Fig. 2 | Manipulation of the PMv^DAT neuron activity allows bidirectional control of maternal aggression. a** Bilateral Cre-dependent ChR2 transduction and fiber implants in the PMv of lactating dams (left) and confocal image of a sample section counterstained with the pan-neuronal marker NeuN, used to validate ChR2 expression and stereotactic implantation of optic fiber (ft) coordinates (right). **b** Schematic of the experimental design used to perform ChR2 PMv^DAT neuron photoactivation in the resident lactating dam during resident-intruder (RI) testing. Adult male and female intruders were used to minimise the levels of maternal aggression expressed in baseline conditions[54,62]. **c** Sample behaviour raster plots at baseline (top) and during ChR2 stimulation (bottom) in a lactating dam during a RI test. **d** Attack duration with and without photostimulation in lactating dams injected with eYFP or ChR2, during the RI test against adult male or adult female intruders (n = 5 mice against male intruders per group, n = 9 mice against female intruders per group, ordinary one-way ANOVA with Tukey's test for correcting for multiple comparisons, P = 0.1401, P = 0.1534, P = 0.9601 and P < 0.0001 respectively). **e** Attack latency with and without photostimulation in lactating dams injected with eYFP or ChR2, during the RI test against adult male or adult female intruders (n = 5 mice against male intruders per group, n = 9 mice against female intruders per group, ordinary one-way ANOVA with Tukey's test for correcting for multiple comparisons, P = 0.7149, P = 0.9468, P = 0.1043 and P = 0.034 respectively). **f** Bilateral Cre-dependent eNpHR3 transduction and fiber implants in the PMv of lactating dams (left) and confocal image of a sample section counterstained with the pan-neuronal marker NeuN, used to validate eNpHR3 expression and stereotactic implantation of optic fiber (ft) coordinates (right). **g** Schematic of the

experimental design used to perform eNpHR3 PMv^DAT neuron photoinhibition studies in the resident lactating dam during RI testing. **h** Sample behaviour raster plots during baseline and eNpHR3 mediated photoinhibition in a lactating dam during a RI test. **i** Quantification of aggression parameters (n = 8 mice per group, two-tailed paired t-test, P < 0.0001, P = 0.0011, and P = 0.0071 respectively). **j** Quantification of the latency to an attack bout following a photoinhibition episode (n = 8 mice per group, two-tailed paired t-test, P = 0.0131). **k** Bilateral Cre-dependent taCasp3 transduction in the PMv of lactating dams (left) and sample confocal image sections following injection of eYFP (middle) vs. taCasp3 (right), used to validate the extent of the PMv^DAT cell lesion. **l** Schematic of the experimental design used to perform RI tests in lactating dams with complete or partial lesion of the PMv^DAT cells. **m** Sample behaviour raster plots in control vs. PMv^DAT neuron lesioned lactating dams during a RI test. **n** Quantification of aggression parameters (n = 34 trials per group, 12 mice per group, two-tailed unpaired t-test, P < 0.0001, P < 0.0001, and P < 0.0001 respectively). All but two mice were tested three times in the RI test. Two mice, one from each group, were tested twice in the RI test. **o** Quantification of PMv^DAT neurons in eYFP and taCasp3 injected mice (n = 24 sections per group, duplicates from 12 mice per group, two-tailed unpaired t-test, P < 0.0001). See also Figs. S1 and S2. All box plots show the median (center line), 25th and 75th percentiles (box bounds), and minima/maxima (whiskers). Exact P values are provided where P > 0.0001; for P ≤ 0.0001, significance is indicated using asterisks (**** for P ≤ 0.0001, *** for P ≤ 0.001, ** for P ≤ 0.01, * for P ≤ 0.05, ns not significant). Source data are provided as Source Data file.

effect depended on a post-synaptic depolarisation, composed of low- and high-voltage components, similar to the Prl-induced current (Fig. 3n).

To investigate whether maternal hormones acting in the PMv are sufficient to trigger aggression in an inherently non-aggressive female, we performed a 28-day localised application of Prl, OT, or both peptide hormones (co-administration) unilaterally into the nucleus of virgin female adult mice. Two weeks after implantation, we performed a panel of tests to investigate changes in aggressive and/or alloparental behaviour, as well as other behavioural features (Fig. S3a). No attacks or changes in the degree of close investigation were observed in the RI test (Fig. S3b). Notably, however, local co-administration of maternal hormones in the PMv decreased sociability in the group (Fig. S3c) and, importantly, impaired alloparenting (Fig. S3d). Modest changes in locomotion, but not anxiety parameters, were identified (Fig. S3e, f).

Together, these experiments indicate that PMv-targeted application of maternal hormones in non-lactating females does not lead to the expression of aggression - suggestive of necessary upstream and/or downstream changes implemented in the brains of lactating dams - but does decrease sociability and pup retrieval performance.

### Activation of PMv^DAT neurons impairs maternal care

Aggression in dams is part of a behavioural program that turns on at parturition and declines by weaning[7,54]. This program includes a coincident initiation of pup care–such as nursing, grooming and crouching–and maternal aggression, aimed to inflict injury and deter infanticidal conspecifics or predators from attacking the pups[7]. In the final experiment, we tested whether pup care and maternal aggression can be triggered by common anatomical substrates, such as the PMv, or whether activation of defined brain loci/circuitry leads to the execution of dedicated/non-overlapping behavioural programs at the expense of other behaviours.

To investigate this issue, we tested the effect of optogenetic activation of PMv^DAT neurons in the absence of an intruder during a pup retrieval test[112,113]. The pup retrieval test was performed with lactating dams expressing ChR2 or eYFP in PMv^DAT neurons (Fig. 4a). Control (eYFP-expressing) dams retrieved all pups within one min of the task initiation during photostimulation (Fig. 4b, d, e, and Video S2). In contrast, photostimulated ChR2-expressing dams exhibited severely impaired pup retrieval behaviour and, in many cases, failed to retrieve any pups over the maximum test period of three min

(Fig. 4b–e, and Video S3). No aggression towards the pups was observed in any of the trials.

These data suggest that activation of these neurons promotes a specialised behavioural program–the one of maternal aggression–at the expense of other maternal behaviours. Thus, even in the absence of a competing intruder stimulus, normal nursing behaviour is inhibited when the PMv^DAT population is activated.

## Discussion

Maternal aggression is a well-established adaptive behaviour across species, remarkable for its sudden onset and reversibility in adulthood[17,114]. The neuronal substrates that underlie the transient aggressive state of the dam have remained elusive. In recent years, key elements of a brain architecture that drives intermale aggression have been identified[33,34,115–118]; some of these loci have also been implicated in the expression of aggression in females[9,117,119]. However, it cannot be automatically inferred that the same neurons drive maternal attacks, as the behaviour is manifested through distinct action patterns[19,57,117]. Importantly, a circuit-based model of maternal aggression needs to account for the transient nature of this trait.

Here we show that PMv^DAT neurons, a key population in orchestrating intermale aggression[35], are typically quiescent in the virgin female but undergo a dramatic shift to an active state during nursing and are required for the expression of maternal aggression. Stimulation of these cells disrupts maternal pup care, even in the absence of overt aggression. The hallmark hormonal agents of lactation, Prl and OT, can powerfully, and through multiple convergent mechanisms, stimulate these cells into discharge mode. These findings suggest a flexible neural framework for hormonally driven generation of adaptive aggression.

Mechanisms have been explored to explain the inherent–purportedly stable–aggression and non-aggression phenotypes of male animals, and one substrate has been found in the electrical excitability of PMv^DAT neurons[34]. But behavioural and neural traits do not remain static over lifetime[56]. The present results show that PMv^DAT cells are relatively hyperpolarised below firing threshold in virgin females, but enter a depolarised state of action potential discharge in nursing dams. It should be noted that the virgin females used for comparison were in the estrus phase; while unlikely in our assessment, it cannot be excluded that subtle changes in PMv^DAT excitability may occur over the estrous cycle. Strikingly, this dichotomy closely mirrors the electrical properties of PMv^DAT neurons in phenotypically

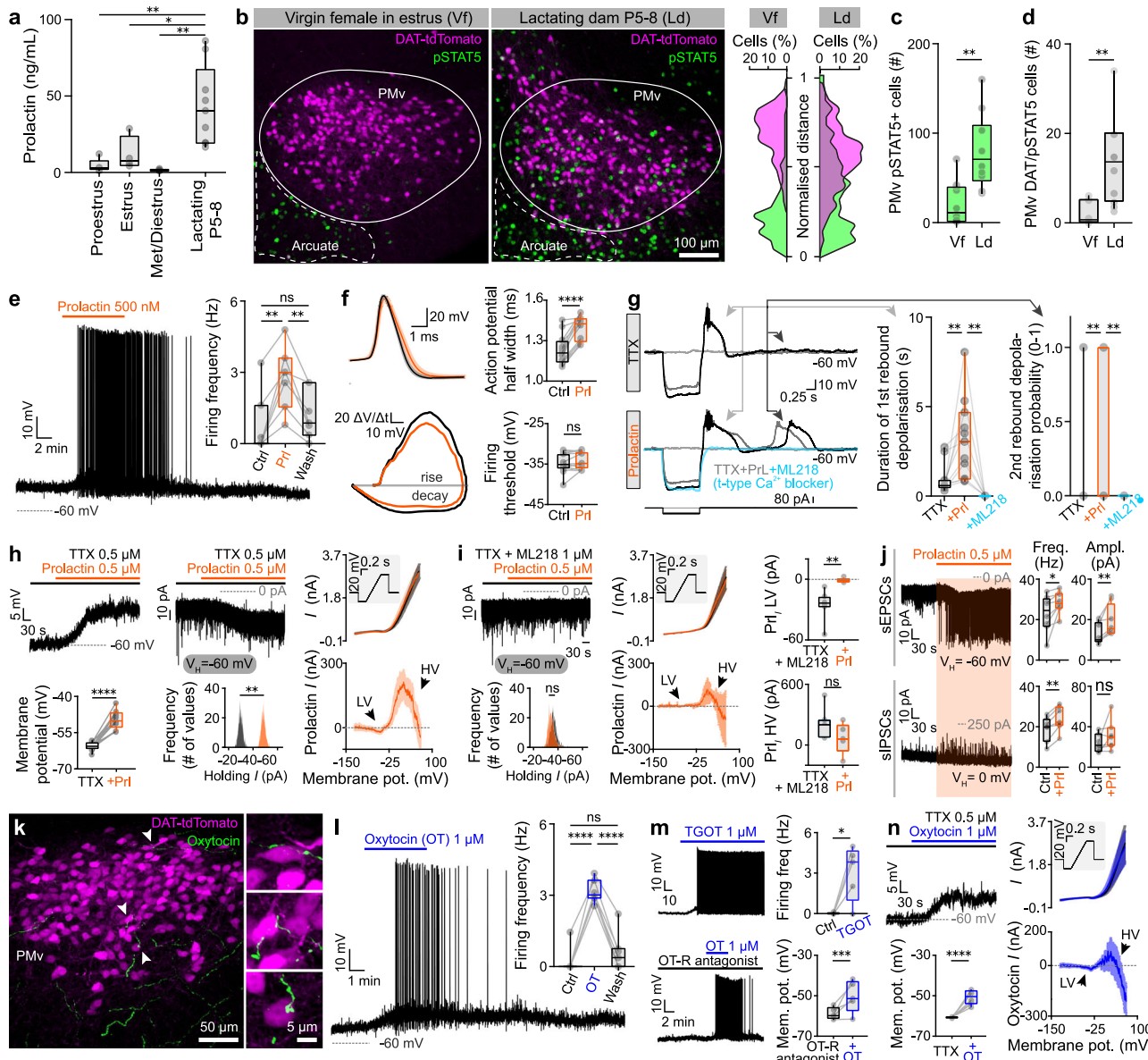

aggressive and non-aggressive males[45]. These findings provide experimental support to the concept that sexual dimorphism in aggression does not result from fundamentally different wiring of the male and female brain (see Beach F. A., 1975[120]). Rather, a common framework, activated by adaptive cues, may underlie situationally relevant attack behaviour in both sexes. This organisation echoes the seminal demonstration that the neuronal substrate for male-like mating behaviour sequences also exists in the female mouse brain[115,117,121]. The present data expand this concept by showing not just the existence of such a "veiled circuit" for maternal aggression in the PMv, but that these neurons are also transiently mobilised when offspring need to be defended to execute attacks until pups have left the nest. It remains to be determined if the different motor expressions of inter-male and maternal aggression[19,57,117] reflect the engagement of distinct downstream motor pattern generators.

Sex steroids underlie the brain adaptations during pregnancy, but *post-partum* maternal-specific physiology and behaviour have been largely attributed to pup cues[122]. Physical interactions with pups trigger hormone release in the dam, which serves as the interface to the brain. This is illustrated by the lactation reflex, where suckling on the nipple triggers the release of OT and Prl from the pituitary gland, driving milk production and ejection, respectively. In parallel, these hormones also

initiate and maintain maternal care[21,72,123]. The induction of pSTAT5 in PMv[DAT] neurons during lactation (present findings) opens the possibility that Prl, and possibly also OT, may play a role in the initial excitability switch, but additional puerperal factors likely play a major role in this transition. Acutely, as demonstrated here in recordings from virgin female PMv[DAT] cells, Prl increases discharge (by bringing them towards or beyond the firing threshold), amplifies the input-output ratio (by increasing action potential duration), and accelerates persistent activity (by promoting rebound bursting). Notably, while the net effect of prolactin is excitatory, application of the hormone increases both excitatory and inhibitory synaptic inputs onto PMv[DAT] neurons, which could reflect a homeostatic mechanism recruited to balance the elicited heightened excitability[124,125]. A similar increase in excitability is induced by OT (possibly acting, at least in part, via vasopressin receptors), indicating a synergistic action of the two hormones. Collectively, this broad program of electrophysiological actions could push PMv[DAT] neurons into the active state that they exhibit in phenotypically aggressive mice ([45]; present findings).

The roles of Prl and OT in maternal aggression remain poorly understood. Although these hormones are most commonly discussed in the context of reinforcing bonds between mother and child, there are also data pointing to their involvement in fighting off a potential

**Fig. 3 | Maternal hormones activate PMv^DAT neurons. a** Serum prolactin levels in virgin females across the estrous cycle (proestrus, estrus, met-/diestrus) *vs.* lactating dams (*n* = 5 mice in proestrus, *n* = 4 mice in estrus, *n* = 5 mice in metestrus/diestrus, and *n* = 9 lactating dams, ordinary one-way ANOVA with Tukey's test for correcting for multiple comparisons, ANOVA P value = 0.0007). **b** Confocal images of pSTAT5 immunoreactivity in PMv, in virgin females in estrus *vs.* lactating dams (left). Overlap of pSTAT5 and DAT-tdTomato fluorescence along the Y axis, with 0 being the most ventral point of the confocal image and 1 the most dorsal (right). **c** Quantification of pSTAT5 immunoreactive cells in the PMv of virgin females in estrus *vs.* lactating dams (*n* = 10 sections per group, 5 mice per group, 2 sections per mouse, two-tailed unpaired t-test, *P* = 0.0011). **d** Quantification of DAT-tdTomato colocalising pSTAT5 immunoreactive cells in the PMv of virgin females in estrus *vs.* lactating dams (*n* = 10 sections per group, 5 mice per group, 2 sections per mouse, two-tailed unpaired t-test, *P* = 0.0015). **e** Application of prolactin leads to a reversible depolarisation of PMv^DAT neurons (*n* = 7 cells from 5 mice, RM one-way ANOVA with Tukey's test for correcting for multiple comparisons, ANOVA *P* value = 0.0016). **f** Prolactin-induced changes in the action potential waveform of PMv^DAT neurons (left). Traces represent mean dV/dt plotted against membrane voltage, computed from 15 action potentials per condition per cell (7 cells from 5 mice). Error bands show mean ± SEM across cells, computed at each voltage bin. Quantification of action potential half-width and firing threshold in control vs. prolactin (black vs. orange, right). Statistical comparisons use two-tailed paired t-test, *P* < 0.0001 and *P* = 0.0746 respectively. **g** *Left* - comparison of post-inhibitory rebound bouts in control (TTX) *vs.* prolactin application (TTX + prolactin) *vs.* co-application with the T-type Ca^2+ channel blocker ML218 (TTX + prolactin + ML218). Black traces represent the hyperpolarisation induced by the highest absolute amplitude negative current pulse, with the effect of a smaller amplitude pulse illustrated in dark gray and the absence of any pulse in light gray. *Middle* - quantification of the 1st post-inhibitory rebound in control (black) *vs.* prolactin (orange, *n* = 11 cells per group collected from 6 mice, two-tailed paired t-test, *P* = 0.0098) and quantification of the 1st post-inhibitory rebound in prolactin (orange) *vs.* ML218 (teal), *n* = 7 cells per group collected from 3 mice, two-tailed paired t-test, *P* = 0.0041). *Right* - quantification of a 2nd post-inhibitory rebound probability in control (black) *vs.* prolactin (orange, *n* = 11 cells per group collected from 6 mice, two-tailed paired t-test, *P* = 0.0061) and quantification of the 1st post-inhibitory rebound in prolactin (orange) *vs.* ML218 (teal), *n* = 7 cells per group collected from 3 mice, two-tailed paired t-test, *P* = 0.0082). **h** Identification of a prolactin-induced postsynaptic depolarisation on PMv^DAT cells in current clamp (left, *n* = 13 cells per group collected from 8 mice, two-tailed paired t-test, *P* < 0.0001) and voltage clamp (middle, *n* = 5 cells per group collected from 5 mice, two-tailed paired t-test, *P* = 0.0078). Characterisation of the

prolactin-induced current across the membrane voltage spectrum in PMv^DAT neurons, with a low- (LV) and high-voltage (HV) component (right, *n* = 7 cells collected from 4 mice). Error bands show mean ± SEM across cells. **i** Application of the selective T-type Ca^2+ current blocker, ML218, blocks the low-voltage component of the prolactin-induced current (Prl_l). Application of prolactin does not induce an inward current in PMv^DAT neurons pre-exposed to ML218 (left, *n* = 6 cells per group collected from 4 mice, two-tailed paired t-test, *P* = 0.5129). In the presence of ML218 the prolactin-induced current is composed only of a high-voltage component (middle, *n* = 7 cells collected from 4 mice). Comparison of the low- (right, *n* = 7 cells in control group [TTX+Prl] and *n* = 5 cells in the ML218 group [TTX+Prl+ML218], two-tailed unpaired t-test, *P* = 0.0042) and high-voltage (right, *n* = 7 cells in control group [TTX+Prl] and *n* = 5 cells in the ML218 group [TTX+Prl+ML218], two-tailed unpaired t-test, *P* = 0.1384) components of the prolactin-induced current in the absence *vs.* presence of ML218. Error bands show mean ± SEM across cells. **j** *Top* - electrophysiology recordings of spontaneous excitatory postsynaptic currents (sEPSCs) during application of gabazine (*n* = 8 cells collected from 4 mice, two-tailed paired t-test, *P* = 0.033 for Frequency [Hz], and *P* = 0.0048 for Amplitude [pA]). *Bottom* - electrophysiology recordings of spontaneous inhibitory post-synaptic currents (sIPSCs, *n* = 7 cells collected from 3 mice, two-tailed paired t-test, *P* = 0.0012 for Frequency [Hz], and P = 0.2009 for Amplitude [pA]). **k** Oxytocin-immunoreactive fibers forming close appositions to PMv^DAT neurons (*n* = 5 mice). **l** Oxytocin leads to a reversible depolarisation of PMv^DAT neurons (*n* = 7 cells per group, collected from 5 mice, RM one-way ANOVA with Tukey's test for correcting for multiple comparisons, *P* < 0.0001, *P* < 0.0001, and *P* = 0.2544 respectively). **m** Application of the selective oxytocin receptor agonist, TGOT, leads to depolarisation of PMv^DAT neurons (top, *n* = 5 cells per group, collected from 5 mice, two-tailed paired t-test, *P* = 0.0274,). Under oxytocin receptor blockade, the oxytocin-induced depolarisation of PMv^DAT neurons persists (bottom, *n* = 5 cells per group, collected from 5 mice, two-tailed paired t-test, *P* = 0.0008). **n** Identification of an oxytocin-induced postsynaptic depolarisation on PMv^DAT cells in current clamp (left, *n* = 8 cells per group collected from 4 mice, two-tailed paired t-test, *P* < 0.0001). Characterisation of the oxytocin-induced current across the membrane voltage spectrum in PMv^DAT neurons, with a low- and high-voltage component (right, *n* = 8 cells collected from 4 mice). Error bands show mean ± SEM across cells. All box plots show the median (center line), 25th and 75th percentiles (box bounds), and minima/maxima (whiskers). Exact P values are provided where *P* > 0.0001; for *P* ≤ 0.0001, significance is indicated using asterisks (**** for *P* ≤ 0.0001, *** for *P* ≤ 0.001, ** for *P* ≤ 0.01, * for *P* ≤ 0.05, *ns* not significant). Source data are provided as Source Data file.

---

threat to offspring[19,20,22–25,126]. A recent study proposed that Prl (acting in the ventrolateral subdivision of the ventromedial hypothalamus [VMHvl]) serves to dampen aggression[127], seemingly at odds with the model suggested by our results. However, the nucleus-specific receptor deletion that was employed in the study may obscure the cumulative effect of the hormone in the brain on behaviour. Indeed, while stimulation of PMv^DAT cells is a powerful inducer of attacks in the puerperal state (present data), chemogenetic activation of VMHvl Prl receptor neurons has no effect on maternal aggression[127]. Our study and Georgescu et al.'s[127] together indicate that Prl may be a powerful modulator of aggression in dams which merits further investigation. Our data further suggest that isolated action of Prl and OT in the PMv of virgin female mice is not sufficient to trigger aggression, but that additional circuit adaptations related to pregnancy and/or nursing are required to establish this behaviour in dams. This idea is also supported by the inability of photoactivating PMv^DAT cells to elicit aggressive behaviour in virgin female mice (Fig. S1c–g) and non-aggressive lactating dams (Fig. S1h, i). It cannot be excluded that there may be circumstances when virgin female mice could mount an attack towards a weaker conspecific in response to exogenous PMv^DAT stimulation, as we did not apply optogenetic stimulation in an encounter towards juvenile intruders, which more effectively triggers aggression in females[9,12]. Even in that scenario, however, the conclusion would remain that the bar to elicit attack has been significantly lowered, if not altogether removed, with the advent of pregnancy. Our findings favour a model where factors linked to pregnancy transform the PMv^DAT cells

into a hyperexcitable state, likely concomitant with reconfiguration of other elements of the aggression circuitry located further downstream, enabling them to translate PMv-driven stimulation into attack behaviour. In this model, with parturition, the surge in Prl and OT provides a potent depolarising influence that may synergise with circuit-level adaptations to drive the full expression of maternal aggression. Future work, including temporally precise pharmacological blockade or genetic disruption of OT and Prl signalling in PMv^DAT cells in dams is necessary to determine whether these hormonal pathways are necessary and sufficient for triggering maternal aggression. Our current findings provide a foundation for such investigations and suggest a broader framework in which hormone-sensitive circuits can be rapidly reconfigured to drive adaptive behavioural transitions.

Finally, we show that stimulation of the PMv^DAT neurons—the same paradigm that powerfully promoted aggressive behaviour in the resident-intruder test—does not elicit pup-directed aggression but markedly impairs parental care during a pup retrieval task. These data firstly suggest (as previously shown for the same population in males[45]) that activation of PMv^DAT cells does not indiscriminately promote attacks against any available target, *i.e.*, pups, but rather may be specific for juvenile and adult intruders. This property of context specificity distinguishes PMv^DAT neurons from the well-characterised VMHvl^ESR1 cells, which drive attacks equally towards conspecifics or inanimate objects[33]. Secondly, it is clear that PMv^DAT neurons do not constitute a hub for driving the full repertoire of maternal behaviours[128]. Rather, an intriguing speculation that these results invite is that parental care of

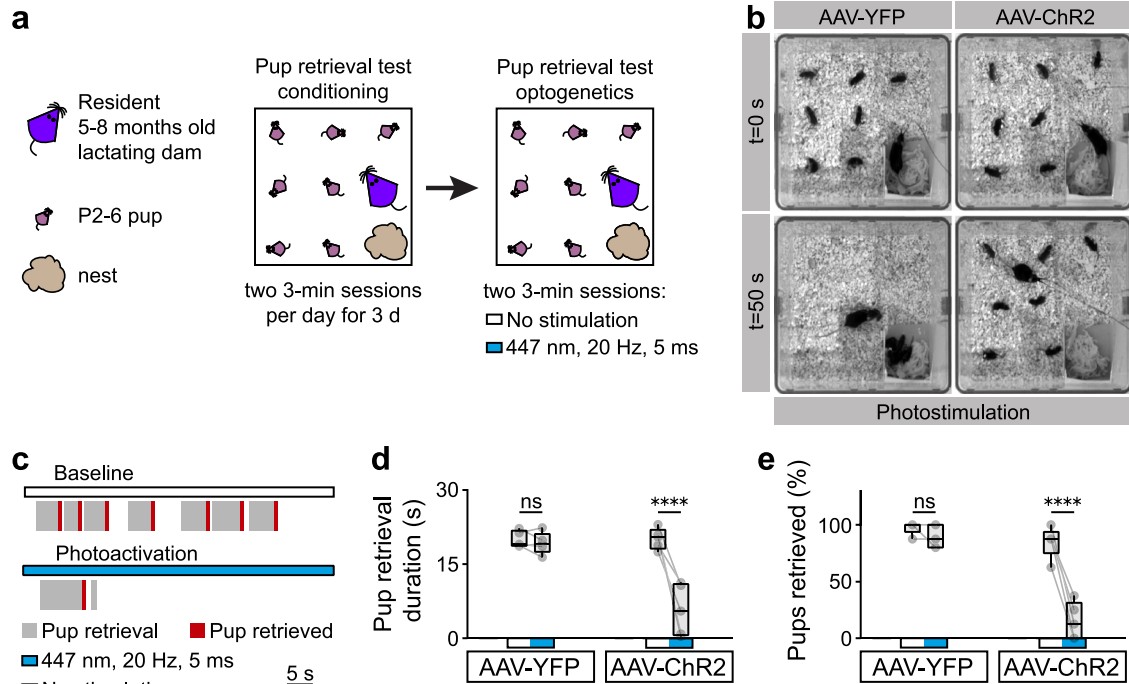

**Fig. 4 | Activation of PMv^DAT neurons suppresses non-aggression-related behavioural programs. a** Schematic of the experimental design used to perform ChR2 PMv^DAT neuron photoactivation during the pup retrieval test. **b** Images from two photostimulation trials at the start (0 s) and 50 s *post*-trial initiation, with lactating dams expressing eYFP (left) and ChR2 (right) in PMv^DAT neurons. **c** Sample behaviour raster plots of two sequential trials without and with photostimulation in a ChR2-expressing lactating dam during the pup retrieval test. **d** Pup retrieval duration with and without photostimulation in lactating dams injected with eYFP or ChR2, during the pup retrieval test ($n = 5$ mice per group, ordinary one-way ANOVA with Tukey's test for correcting for multiple comparisons, $P > 0.9999$ and

$P < 0.0001$ respectively). **e** Percentage of pups retrieved with and without photostimulation in lactating dams injected with eYFP or ChR2, during the pup retrieval test ($n = 5$ mice per group, ordinary one-way ANOVA with Tukey's test for correcting for multiple comparisons, $P = 0.9820$ and $P < 0.0001$ respectively). All box plots show the median (center line), 25th and 75th percentiles (box bounds), and minima/maxima (whiskers). Exact $P$ values are provided where $P > 0.0001$; for $P \leq 0.0001$, significance is indicated using asterisks (**** for $P \leq 0.0001$, *** for $P \leq 0.001$, ** for $P \leq 0.01$, * for $P \leq 0.05$, ns not significant). Source data are provided as Source Data file.

offspring is actively suppressed during aggression orchestrated via the PMv^DAT population. Indeed, in virgin females the infusion of OT and Prl into the PMv resulted in a disruption of alloparenting (Fig. S3). It should be noted, however, that the impairment of pup care following optogenetic PMv^DAT activation does not automatically mean that this system plays a physiological role in suppressing nursing, but may impact the prioritisation of maternal behaviours; future work will need to explore this aspect in greater depth.

The data presented here suggest that the transient activation of a circuit can enable an animal to reversibly access a behaviour for adaptive purposes. Our results suggest a model in which PMv^DAT neurons are hormonally primed into a hyperexcitable state in the lactating dam, greatly lowering the threshold for eliciting attacks against conspecifics. With the sudden appearance of an intruder, activation of these neurons may serve to prioritise attack over nursing to direct attention and resources to the most imminent threat to the pups' survival.

## Methods
### Animals
All animal experiments had received approval from the local ethical board, *Stockholms Norra Djurförsöksetiska Nämnd*, and were performed in accordance with the European Communities Council Directive of November 24, 1986 (86/609/EEC). Wild-type mice with C57BL/6 J and BALB/c background were used, in addition to previously generated C57BL/6 J *Slc6a*^Cre (DAT-Cre) knock-in[129] and floxed-tdTomato mice (The Jackson Laboratory, strain datasheet–007909). Animals were group-housed, up to five per cage, in a temperature

(23 °C) and humidity (55%) controlled environment, in a 12 h light, 12 h dark cycle with *ad libitum* access to food and water. Cages were changed on a weekly basis.

Adult virgin female, and lactating female, mice used in experiments were single housed for a period of two to four weeks, depending on the experimental design. Only multiparous lactating mice were used in the experimental group of *lactating dams*, typically following their third or fourth parturition.

### Viral vectors
For channelrhodopsin optogenetic studies animals were injected in the PMv with 400 nL of AAV5-EF1a-DIO-hChR2(H134R)-eYFP-WPRE-hGH (Addgene20298) $8.41 \times 10^{12}$ genomic copies per mL. For halorhodopsin-mediated neuronal silencing, animals were injected with 400 nL of AAV5-EF1a-DIO-eNpHR3.0-eYFP-WPRE-hGH (Addgene26966) $7.02 \times 10^{12}$ genomic copies per mL. The optogenetic control groups were injected with 400 nL of AAV5-EF1a-DIO-eYFP-WPRE-hGH (Addgene27056) $5.82 \times 10^{12}$ genomic copies per mL. The ChR2, eNpHR3 and eYFP AAV5 were prepared by the University of Pennsylvania Vector Core. For targeted cell ablation of PMv^DAT neurons animals were injected in PMv with 300 nL of AAV5-flex-taCasp3-TEVp $2.9 \times 10^{12}$ genomic copies per mL, and the AAV was prepared by the viral vector core at the University of North Carolina. Viral injections were performed bilaterally.

### Stereotactic surgery and viral gene transfer
Adult DAT-Cre mice 5–8 months old (sexually inexperienced) were stereotactically injected with a virus, and implanted with fiber implants

when purposed for in vivo optogenetic behaviour experiments, and were individually housed for 2 weeks post-surgery. Animals were anaesthetised with isoflurane (1–5%) and placed in a stereotactic frame (David Kopf Instruments). Virus was injected into the PMv bilaterally using a pulled glass capillary (World Precision Instruments) by nano-litre pressure injection at a flow rate of 50 nL per min (Micro4 controller, World Precision Instruments; Nanojector II, Drummond Scientific). Stereotactic injection coordinates to target the PMv were obtained from the Paxinos and Franklin atlas[130] (Bregma: −2.45 mm, midline ±0.6 mm, dorsal surface −5.5 mm). Ferrules and fiber-optic patch cords were purchased from Thorlabs and Doric Lenses, respectively. The virus-injected animals were housed individually during a 2-week recovery period, and then examined behaviourally and histologically.

In the case of the *lactating dams* group, sexually experienced female mice with prior maternal experience were placed in a cage with a male mouse, and were checked for plugs in the morning hours of the next four days[131]. Following identification of plugs, female mice were single housed, and surgeries were performed for AAV injections, and/or optic fiber implants, as above. The surgical procedures occurred within a week of plug identification. Female mice in which plug identification did not occur or was ambiguous were excluded from the study.

## Optogenetics

In optogenetic experiments, subjects were coupled via a ferrule patch cord to a ferrule on the head of the mouse using a zirconia split sleeve (Doric Lenses). The optical fiber was connected to a laser (447 nm for ChR2; 635 nm for eNpHR3; CNI-MLL-III-447-200-5-LED and CNI-MLL-III-635-200-5-LED, CNI lasers 200 mW) via a fiber-optic rotary joint (FRJ_1×1_FC-FC, Doric Lenses) to avoid twisting of the cable caused by the animal's movement. After a testing session, DAT-Cre animals were uncoupled from the fiber-optic cable and returned to a housing room. The frequency and duration of photostimulation were controlled using custom written LabView software. Laser power was controlled by dialling an analogue knob on the power supply of the laser sources. Light power was measured from the tip of the ferrule in the patch cord before being installed in the brain (the ferrule-connector system) at different laser output settings, using an optical power and energy meter and a photodiode power sensor (Thorlabs). Upon identification of the fiber placement coordinates in brain tissue slides, irradiance (light intensity) was calculated using the brain tissue light transmission calculator based on (http://www.stanford.edu/group/dlab/cgi-bin/graph/chart.php) using laser power measured at the tip and the distance from the tip to the target brain region measured by histology. Animals showing no detectable viral expression in the target region and/or ectopic fiber placement were excluded from analysis. In some experiments, animals were photostimulated with a train of 473 nm light (20 Hz, 5 msec, 5 min) 45 min before perfusion in the absence of an intruder at an intensity which had evoked a behavioural phenotype in the final testing session. Brain sections were subsequently immunohistochemically labelled for c-Fos to identify optogenetically activated cells.

## Behavioural tests

Behavioural tests were performed at 2 h post-initiation of the light phase and 2 h prior to initiation of the dark phase. Mice were acclimated to the testing facility for 1 h prior to testing. Behaviours were recorded using a digital video recording unit. Behavioural annotations were performed manually, with conditions blinded to the experimenter performing the behavioural scoring.

## Resident-intruder (RI) test

RI tests were initiated at 2–4 weeks post-surgery and repeated weekly for 2–5 weeks. All dams used in the present study were multiparous and were used in behavioural experiments during their third or fourth litter. Mouse cages were not cleaned for a minimum of 3 days prior to the behavioural test. Intruders were individually introduced to a DAT-Cre mouse in a testing session in a random order with respect to sex, with a 5 min interval between intruders. The types of intruders were: juvenile females, adult females and adult males. Subjects involved in the RI test were exposed to 3 experimental days of the test, resulting in multiple measurements per resident.

## ChR2-mediated activation in RI

After the introduction of an intruder, a virus-injected animal was recorded for 3 min to assess baseline behaviour towards each intruder. The baseline recording was followed by photostimulation trials (3 min in duration) with varying irradiance (intensity), stimulation intervals, distance, orientation and recent behaviour history between the two animals at the onset of photostimulation. The AAV5-DIO-eYFP and AAV5-DIO-ChR2 injected experimental animals were processed in random order.

## eNpHR3-mediated silencing in RI

DAT-Cre lactating females expressing eNpHR3 or control eYFP were introduced to between one to four juvenile female intruders, in three acclimation sessions without photostimulation to assess baseline aggression as well as to augment aggressiveness. Testing sessions initiated with recordings in the absence of laser stimulation to assess baseline behaviour towards each intruder on the day of the experiment. In photostimulation trials, irradiance (intensity) ranging from 2.1–11.6 mW/mm² was delivered continuously in varying intervals (typically 2–10 sec) depending on the phase of the resident-intruder interaction. To examine whether photostimulation during charging stopped escalation to attack, residents were photostimulated during body realignment to gain access to the back of the intruder.

## Pup retrieval test

Adult virgin female or lactating mice 5–8 months old were exposed to foster pups or to their own litter, respectively. The litter size per test is described in the individual tests/figures. Material for nest building was provided in each cage. The activity in the cage was video recorded, and maternal behaviours were scored by an experimenter blind to the experimental conditions.

## Three-chambered sociability test

The social approach apparatus was an open-topped box made of acrylic (60 cm in length × 40 cm in width × 20 cm in height), and divided into three chambers with two clear acrylic walls. Dividing walls had retractable doorways allowing access into each chamber. The wire cup used to contain the novel mice was made of cylindrical chrome bars spaced 1 cm apart (10 cm H; bottom diameter: 10 cm). Test mice were confined in the center chamber at the beginning of each phase. Exploration of an enclosed mouse or an empty wire cup was defined as when a test mouse oriented toward the cup with the distance between the nose and the cup less than 1 cm, or as climbing on the cup. The time spent in each chamber and time spent exploring enclosed novel mice or empty cups (the novel objects) were recorded from an overhead camera, and analysed using Ethovision XT12.

## Open field test (OFT)

OFTs were performed in a white acrylic box (45 × 45 × 40 cm) with an overhead lamp directed to the centre of the field, providing 120 lux of illumination on the floor of the arena. Each mouse was placed in the corner of the apparatus and locomotion parameters were recorded for 20 min.

## Elevated plus maze (EPM)

The EPM test was performed using a polyvinyl chloride maze comprising a central part (5 × 5 cm), two opposing open arms (32.5 × 5 cm),

and two opposing closed arms ($32.5 \times 5 \times 32.5$ cm). The apparatus was set to a height of 50 cm, and the open arms were provided with uniform overhead illumination of 6 lux. Mice were placed in the open arm point close to the centre facing the closed arms, and video recordings were performed for a total duration of 20 min.

## Chronic administration of maternal hormones in the PMv

For localised chronic administration of maternal hormones in the PMv, we performed cannula implantations using the Paxinos and Franklin atlas[130] coordinates (Bregma: −2.45 mm, midline ±0.6 mm, dorsal surface −5.5 mm). Cannulae (0008663 Brain Infusion Kit 2) and osmotic minipumps (0000298 ALZET Model 2004) were purchased from ALZET, and prepared according to protocol.

## Brain slice electrophysiology

Acute slices of the mediobasal hypothalamus were prepared from adult DAT-tdTomato mice (own breeding). Mice were deeply anaesthetised with sodium pentobarbital (200 mg/kg, i.p., Sanofi-Aventis, France) and euthanised by decapitation prior to brain extraction for slice preparation, in accordance with institutional and national ethical guidelines. Slices were cut on a vibratome (Leica VT1000S) to 250 μm thickness and continuously perfused with oxygenated aCSF containing (in millimolar): NaCl (127), KCl (2.0), NaH$_2$PO$_4$ (1.2), NaHCO$_3$ (26), MgCl$_2$ (1.3), CaCl$_2$ (2.4), and D-glucose (10), at near-physiological temperature ($33 \pm 1$ °C) during recording. Each slice was exposed only to a single bath application of pharmacological compounds and was used for a single experiment. Whole-cell current- and voltage-clamp recordings were performed with micropipettes filled with intracellular solution containing (in millimolar), K-gluconate (140), KCl (10), HEPES (10), EGTA (10), and Na$_2$ATP (2) (pH 7.3 with KOH). Recordings were performed using a Multiclamp 700B amplifier, a DigiData 1440 digitiser, and pClamp 10.2 software (Molecular Devices). Slow and fast capacitative components were semi-automatically compensated. Access resistance was monitored throughout the experiments, and neurons in which the series resistance exceeded 15 MΩ or changed ≥20% were excluded from the statistics. Liquid junction potential was 16.4 mV and not compensated. The recorded current was sampled at 20 kHz.

Reagents used in slice electrophysiology experiments; Neurobiotin™ tracer (Vector laboratories) was used in combination with Streptavidin, DyLight™ 405 conjugated (21831 Thermoscientific) or Avidin-FITC (43–4411 Invitrogen). TTX was purchased from Alomone Labs. Oxytocin nonapeptide, the OT-receptor (OT-R) agonist, (Thr4, Gly7)-oxytocin (TGOT), and the OT-R antagonist, vasotocin [d(CH2)$_5$$^1$, Tyr(Me)$^2$, Thr$^4$, Orn$^8$, des-Gly NH$_2$$^9$], were purchased from Bachem. Prolactin CYT-321 was purchased from PROSPEC, and SR 95531 (Gabazine) and ML218 hydrochloride were purchased from Tocris. OriginPro9 was used for electrophysiological data analysis.

## Immunofluorescence

Mice were anaesthetised with sodium pentobarbital (200 mg/kg, i.p., Sanofi-Aventis), then transcardially perfused with 10 mL Ca$^{2+}$-free Tyrode's solution (37 °C) containing 0.2% heparin, followed by 10 mL fixative (4% paraformaldehyde and 0.4% picric acid in 0.16 M phosphate buffer (PBS), 37 °C), then 50 mL ice cold fixative. Whole brains were dissected, immersed in ice cold fixative for 90 min then stored in 0.1 M PBS (pH 7.4) containing 20% sucrose, 0.02% bacitracin and 0.01% sodium azide for three days, before freezing with CO$_2$. Coronal sections were cut at a thickness of 14 μm on a cryostat (Microm) and thaw-mounted onto gelatin-coated glass slides. For indirect immunofluorescence staining (performed at room temperature unless otherwise specified), air-dried sections were washed in 0.01 M PBS for 30 min before incubation with primary antisera diluted in PBS containing 0.3% Triton X-100 and 1% BSA for 16 h at 4 °C. The slides were then washed for 30 min in PBS followed by 2 h incubation with Alexa-488-conjugated donkey anti-rabbit secondary antisera (1:500;

Invitrogen). Slides went through a final wash for 30 min in PBS and mounted with glycerol containing 2.5% DABCO; Sigma. This method was used with the following antibodies: NeuN was detected with primary antibody rabbit anti-NeuN (1:500; Cell Signaling, D4G40), eYFP was detected with chicken anti-GFP (1:500; Aves Labs, GFP-1020). Immunohistochemistry for pSTAT5: Prior to immunofluorescence staining, antigen retrieval was performed by incubating sections for 15 min in citric acid (pH 7.4) at 80 °C, then cooled at room temperature for a further 30 min. After a 1% H$_2$O$_2$ Tris-buffered wash, sections were incubated in rabbit pSTAT5 primary antibody (pSTAT5 Tyr 694, Cat#: C11C5, 1:500; Cell Signaling Technology) for 72 h at 4 °C. Primary antibody incubation was followed by Alexa-488-conjugated donkey anti-rabbit secondary antisera (1:500; Invitrogen).

To perform c-Fos immunostaining, mice were deeply anaesthetised with sodium pentobarbital (as described above) and perfused transcardially with 4% (weight/vol) ice-cold paraformaldehyde in 0.1 M PBS. Brains were post-fixed in the same solution and 40 μm-thick coronal slices were cut at the vibratome (Leica). Two PMv coronal sections per animal were selected to perform c-Fos immunostaining as follows. Sections were washed in TBS (100 mM Tris-Cl, 150 mM NaCl, pH 7.5), incubated 1 h at 25 °C in 1% BSA-0.3% Triton X-100-TBS solution and then kept at 4 °C in rabbit anti-c-Fos antibody (sc-52 LotG1108, Santa Cruz Biotechnology) solution (1:200 in 1% BSA-TBS) overnight. After TBS washing, sections were incubated for 1 h at 25 °C in Alexa Fluor® 647 (Invitrogen) goat anti-rabbit secondary antibody (1:500 in 1% BSA-TBS).

## Microscopy and cell-counting

All brain slices were imaged by epifluorescence microscopy (ZEISS Imager M1) or confocal microscopy (Zeiss, LSM 800) for subsequent analysis. Brain areas were determined according to their anatomy using the Paxinos and Franklin Brain Atlas[130]. For PMv$^{DAT}$ cell counts the entire PMv was cut, stained and counted. Quantification of c-Fos staining was obtained by averaging the number of positive cells of right and left PMv in two brain sections (−2.46 and −2.54 mm from Bregma[130]). All counts were performed manually using Image J software and the annotator was blind to test conditions.

## Randomisation and blinding

Behavioural data collection and analysis was performed blind to experimental conditions. Anatomy data analysis but not tissue collection was blinded. Electrophysiological data sampling and analysis was not blinded, with the exception of all whole-cell patch clamp datasets presented in Fig. 1. Mice were first screened for expressing maternal aggression and then further assigned to groups for behavioural experiments.

## Statistical analysis

No statistical methods were used to pre-determine sample sizes but our sample sizes are similar to those reported in previous publications[2,11,14,30,31]. Data met the assumptions of the statistical tests used, and were tested for normality and equal variance. All $t$-tests and one-way ANOVAs were performed using Graph Pad Prism software (Graphpad Software Inc.). The Tukey and Bonferroni *posthoc* tests were used as appropriate for one-way ANOVAs. Normality was determined by D'Agostino–Pearson normality test. Statistical significance was set at $P < 0.05$.

## Reporting summary

Further information on research design is available in the Nature Portfolio Reporting Summary linked to this article.

# Data availability

Data supporting the conclusions of the present study are available at the following Zenodo link. Source data are provided with this paper.

## Code availability

Analysis was performed with proprietary software specified in the "Methods" sections.

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

## Acknowledgements
Members of the Broberger laboratory are thanked for advice and discussion during the preparation of this manuscript, as are the many investigators who originally developed the reagents and tools that made this study possible. The authors gratefully acknowledge support by a project grant from the Knut and Alice Wallenberg Foundation (2020.0054), a European Research Council Advanced Grant (TOGE-THER 101021496), a Distinguished Professor Grant from the Swedish Research Council (021-00671), funding from *Hjärnfonden* (the Swedish Brain Foundation), and internal Stockholm University funds to C.B. S.S. received support from the Wenner-Gren Foundations (WGF) and the Swedish Society for Medical Research (SSMF).

## Author contributions
S.S. and C.B. conceived the study and wrote the manuscript. S.S. designed, performed, and analysed experiments. P.W., G.S., G.F., and C.B. designed experiments. P.W., G.S., S.K., K.Z., and L.H. performed experiments. All authors reviewed the manuscript.

## Funding

## Competing interests
The authors declare no competing interests.
