## [Transparent Peer Review file · Nature Communications]

Maternal Aggression Driven by the Transient Mobilisation of a Dormant Hormone-Sensitive Circuit

Corresponding Author: Dr Christian Broberger

Version 0:

Reviewer comments:

Reviewer #1

(Remarks to the Author)

The reviewer appreciates the effort made by the authors to improve the study. The results showing differential excitability of aggressive and non-aggressive lactating females are informative, further supporting a relationship between PMv excitability and aggression. The experiments shown in the paper are done elegantly, and the data quality is high. However, the manuscript still feels composed of two disconnected parts. Figures 1 and 2 show that PMv excitability is central to maternal aggression, while Figures 3 and 4 show PMv cells can be modulated by oxytocin and prolactin, but these hormones appear to play no role in aggression; instead, they suppress parental behaviors and cause behavioral changes in open field and elevated plus maze. The reviewer believes that it is crucial to tie these two parts together. Specifically, the study will benefit from a functional experiment that demonstrates the role of oxytocin/prolactin in maternal aggression. For example, does OXTR antagonist injection into the PMv suppress maternal aggression in lactating moms? Does OXTR or Plr KO at the PMv (a strategy used by the group in previous studies) impair maternal aggression? While the authors showed that oxytocin and prolactin do not increase aggression in virgin females, it remains possible that the hormones will boost aggression in lactation moms. If such a link really could not be found, the authors may consider splitting Figures 1&2 and 3&4 into two papers since they do not support one coherent conclusion. Lastly, several places state "Data not shown", please show those data to support the statement.

Reviewer #4

(Remarks to the Author)

NCOMMS-24-19332-T: Maternal Aggression Driven by the Transient Mobilization of a Dormant Hormone-Sensitive Circuit

Overall:

In the manuscript "NCOMMS-24-19332-T: Maternal Aggression Driven by the Transient Mobilization of a Dormant Hormone-Sensitive Circuit" the authors attempt to dissect the neuronal circuits and the brain hubs regulating maternal aggression in mice. They found out that PMVDAT neurons are activated and seem to be needed during the expression of maternal aggression. Additionally, their ex-vivo data portray a picture suggesting that the hormones oxytocin (OT) and prolactin (PrL), which are known to be upregulated during lactation, are affecting PMVDAT neurons' excitability and may contribute to the PMVDAT regulation of maternal aggression. The manuscript is well-written, the panels and figures are well-displayed and the results seem robust. Although the manuscript has already undergone one round of revisions and the authors provide explanations as well as consistent changes in the text, this reviewer has found some major points which should be addressed before publication.

Introduction, abstract, and general

- In the abstract the authors write "Aggression, a sexually dimorphic behaviour, is prevalent in males and typically absent in virgin females". That definitely might be the case for C57BL6 strains but it is definitely not the case for other species such as humans, rats, hamsters, other mouse strains, *Peromyscus*, and many others. The authors should be careful to not limit their view of virgin female aggression research to mouse models. Indeed, several studies in the last years have used other

species to access virgin female aggression. It comes to my attention that those studies have not been cited by the authors, that feels a bit troubling, especially taking into account that research done in females already gets fewer citations than research performed in males. Among the manuscripts is important to highlight:

Oliveira VEM, Lukas M, Wolf HN, Durante E, Lorenz A, Mayer AL, Bludau A, Bosch OJ, Grinevich V, Egger V, de Jong TR, Neumann ID. Oxytocin and vasopressin within the ventral and dorsal lateral septum modulate aggression in female rats. *Nat Commun.* 2021 May 18;12(1):2900. doi: 10.1038/s41467-021-23064-5. PMID: 34006875; PMCID: PMC8131389.

Oliveira VEM, Bakker J. Neuroendocrine regulation of female aggression. *Front Endocrinol (Lausanne).* 2022 Aug 10;13:957114. doi: 10.3389/fendo.2022.957114. PMID: 36034455; PMCID: PMC9399833.

Newman EL, Covington HE 3rd, Suh J, Bicakci MB, Ressler KJ, DeBold JF, Miczek KA. Fighting Females: Neural and Behavioral Consequences of Social Defeat Stress in Female Mice. *Biol Psychiatry.* 2019 Nov 1;86(9):657-668. doi: 10.1016/j.biopsych.2019.05.005. Epub 2019 May 13. PMID: 31255250; PMCID: PMC6788975.

Terranova JI, Song Z, Larkin TE 2nd, Hardcastle N, Norvelle A, Riaz A, Albers HE. Serotonin and arginine-vasopressin mediate sex differences in the regulation of dominance and aggression by the social brain. *Proc Natl Acad Sci U S A.* 2016 Nov 15;113(46):13233-13238. doi: 10.1073/pnas.1610446113. Epub 2016 Nov 2. PMID: 27807133; PMCID: PMC5135349.

Silva AL, Fry WHD, Sweeney C, Trainor BC. Effects of photoperiod and experience on aggressive behavior in female California mice. *Behav Brain Res* (2010) 208:528–34. doi: 10.1016/j.bbr.2009.12.038

- On that same aspect the authors write in the abstract “While maternal hormones are known to elicit nursing, their potential role in maternal aggression remains elusive”. That sentence is misleading and undermines the work of researchers such as Prof. Oliver Bosch, Prof. Inga Neumann, and Prof. Benjamin C. Nephew who have worked for the last decades investigating how 1) peptide signaling, 2) peptide release and 3) receptor binding densities affect maternal behaviors including maternal aggression. Although the authors cite some of the reviews written by those researchers, it would be important to give credit to original papers and of course, change this sentence in the abstract and make corrections throughout the manuscript.
- My third general point is regarding the translational relevance of maternal aggression. As an animal researcher myself, I do understand that scientists have to work with the models they have at hand. Also, every animal model has its gains and pitfalls. Although, maternal aggression is an ecologically relevant model for understanding the motivational aspects underlying aggression in rodents that might not be the case for other species including humans. Women exhibit aggression outside of parturition/lactation/pregnancy. Additionally, pregnancy and puerperium have not been tied to the prevalence of aggression disorders in humans. Taking this scenario into account this reviewer invites the authors to critically look into the use of maternal aggression to study the neurobiology of aggression and sex differences in general, as the authors themselves wrote, lactation is a unique physiological period for females in which several systems are upregulated. Thus, one should bear in mind that the maternal brain and behavior may not completely reflect the female brain and therefore cannot directly be compared to the male brain and behavior (aggression). Finally, this point does not aim to undermine lactation as a powerful scientific tool but rather highlights how important this model is to understand the maternal brain, which does not necessarily reflect the non-lactating female brain which is the target of sex differences research. The reviewer would like to know the thoughts of the authors on the topic and have that addressed in the discussion.
- In the introduction the authors write “Importantly, it is not well understood if the neural modules that lead to the expression of aggression are shared between the two sexes, or exhibit sexual dimorphism.” In fact, there are not so many studies in females and on sex differences, that is another reason why the authors should cite studies such as Terranova 2016, Newmann, 2020, and Oliveira, 2021 which actually address this question.

Results, methods, discussion

- Control experiments in virgin females are normally performed in estrus females. What is the rationale for that? Why was this specific phase of the estrous cycle chosen? Especially taking into account that prolactin is released during the evening of proestrus which antecedes estrus. Additionally, the work of Alexa Veenema shows that females in estrus exhibit changes in OXTR densities in hypothalamic regions. Could those factors have influenced the behavioral and ex vivo results of the authors? Why not use diestrus females? Finally, regarding the behavioral perspective, Oliveira 2021 has found that optogenetic stimulation of OXT neurons in the PVN and SON as well as OXT terminals in the lateral septum of virgin female rats failed to elicit aggression when females were either in proestrus or estrus but triggered aggression when females were in metestrus or diestrus. Moreover, estrus has been shown to decrease aggression in virgin females in a couple of studies (to a variable extent). Could direct optogenetic stimulation trigger aggression in diestrus females in the present study? Could the lack of effect reflect an estrous cycle interaction?
- On the same topic, this reviewer noticed that the authors have used juvenile females as intruders for the loss-of-function experiments for lactating females (maternal aggression) in order to enable optimal (high) aggression levels but adult intruders for virgin females. Why did the authors choose this approach? Why not use juveniles for both protocols? Especially knowing that Hashikawa, 2017 and Oliveira, 2019 have shown that virgin female mice and rats, respectively, preferentially attack juveniles. What would be the outcome of performing aggression training using juvenile females as intruders? Would that lead to aggression and activation of PMVDAT neurons?
- Regarding behavioral outcome how do loss- and gain-of-function experiments affect other behavioral domains such as

non-aggressive social investigation and home cage exploration? This data might be relevant, especially knowing that activation of PMVDAT neurons does not seem to be aggression- but rather social investigation-specific.

- Going back to the previous works of the authors on winning experience, the authors have shown that testosterone could generate the needed plasticity to induce winning in males. In females, it does not seem to be so simple as treatment with OT and prolactin chronically failed to affect aggression. However, would direct stimulation of PMVDAT of non-aggressive lactating mice turn those females into aggressors? In the same regard, could direct stimulation of PMVDAT neurons lead to maternal aggression even after OT and prolactin receptor antagonist treatments? These experiments could provide evidence as to whether PMVDAT neurons are downstream OT and Prl. Also were there differences in prolactin and OT levels between aggressors and non-aggressors lactating females?
- Concerning the ex vivo patch-clamp data, I know that the authors were asked by the other reviewers about what would be the effect of treating lactating females with prolactin and oxytocin, to what they answered: "The recordings were performed on slices from virgin female mice, as our hypothesis is not that PMvDAT cells become sensitive to Prl and OT only after giving birth, but rather that the dramatic increase in circulating levels of these hormones makes their action all the more potent. The reason for not using dams for these recordings was the risk that sustained exposure to high concentrations of Prl and OT may have rendered them insensitive to the exogenous applications of these hormones, which would have confounded the results. Adaptation is, in our experience, common in electrophysiological experiments using peptides/polypeptide ligands, and the high degree of STAT5 phosphorylation in PMvDAT neurons in lactating dams (present data) strongly indicates that they are under substantial Prl receptor activation. It was, however, not clear in the original submission that these experiments were performed on tissue from virgin female mice. We apologize for this missing information and have now specified it in the Results." I disagree with the author's point indeed lactating females show increased levels of prolactin and oxytocin but only up to stimulation (suckling). That does not mean those neurons will be hypersensitive to OT or Prl and that the levels of the hormones will be high in the brain during the whole time. Can the authors actually provide literature references for that? Additionally, the authors missed the opportunity to generate the missing link between peptide signaling and excitability changes by treating lactating female slices with OT receptor and prolactin receptor antagonists alone and in combination to investigate whether this could reverse the plasticity changes seen in the lactating brain.
- Also regarding ephys data the authors find an increase iPSCs after prolactin treatment, how do the authors think this data adds to the idea of prolactin turning PMVDAT neurons more excitable?
- Cross-talk between OT and vasopressin receptors has been shown including in the context of female aggression (Oliveira 2021, Tan 2020). To the comment of the other reviewers, the authors claim "Thus, although available histochemical data indicate the expression of OT-R's, but not vasopressin receptors in the PMv these pharmacological results suggest that vasopressin receptors may also contribute to the peptide's action on PMvDAT neurons". Here the authors use a rat reference but they should keep in mind that the patterns of OT and VP receptor expression differ between mice and rats. Meaning that mice could express receptors in different locations and densities than rats. On that aspect, using a V1aR antagonist plus OT treatment could help to elucidate the non-OT receptor-mediated mechanisms described in the manuscript. Additionally, vasopressin itself has been shown to also modulate maternal aggression in rats, could the authors speculate whether vasopressin acts synergically with OT and Prl in their model?

Version 1:

Reviewer comments:

Reviewer #1

(Remarks to the Author)

The reviewer wished that the authors could have addressed the PMv OXTR and PRLR role in maternal aggression more thoroughly but recognized that this requires substantial more effort. While a lot more could be done to fill the gaps in the paper, the reviewer will not request more experiments that will delay the publication of the study. The reviewer hopes that the authors will continue their effort to further elucidate the functional importance of PMv OXT and PRL signaling in maternal aggression.

Reviewer #4

(Remarks to the Author)

First of all, I would like to thank the authors for i) carefully answering all my points, ii) performing one extra experiment in the non-aggressor lactating females, which does show that PMVDAT neuron excitation is not sufficient to trigger maternal aggression but rather that some contextual changes, presumably high levels of OXT and Prl, are needed in combination to stimulate maternal aggression and iii) for assessing the role PMVDAT in social preferences. I acknowledge that this is an extensive amount of extra work and that many points are now clearer. However, I still have one major concern regarding the link between hormones, PMVDAT neurons, and maternal aggression.

I did read the answer of authors to Referee #1, and I can completely relate to the fact that performing extra experiments to connect OT and Prl to the PMVDAT neurons can be extremely time-consuming and might even raise further questions.

Additionally, I also understand the point the authors raised regarding my question about treating lactating female slices with PrL and OT.

However, the title of the manuscript itself as well as the discussion imply a pivotal role of both hormones in the effects reported by the authors in maternal aggression. Please see below:

“Maternal Aggression Driven by the Transient Mobilisation of a Dormant Hormone-Sensitive Circuit”

“With parturition, the surge in Prl and OT provide a powerful depolarising influence synergising with this primed state to push the circuit into hyperactivity and thus drive the full expression of aggression. Such a role for Prl and OT in maternal aggression would also account for the disappearance of this behaviour with weaning, when serum levels of these hormones quickly drop to pre-puerperal levels”

“Our results suggest a model in which PMvDAT neurons are hormonally primed into a hyperexcitable state in the lactating dam, greatly lowering the threshold for eliciting attacks against conspecifics.”

“The hallmark hormonal agents of lactation, Prl and OT, can powerfully, and through multiple convergent mechanisms, stimulate these cells into discharge mode. These findings suggest a flexible neural framework for hormonally driven generation of adaptive aggression”

Although the authors claim in their answer to Referee 1# “The purpose of the experiments in the study that involve these hormones was, however, to connect our findings with the endocrine agents that envelop the maternal state, rather than an exhaustive study of these hormones” they do use both hormones to speculate on their findings without providing data that sustain those statements. This is misleading, taking into account that those experiments were performed in different groups (hormonal in vitro findings as well as the chronic peptide treatments in virgins, whereas behavioral effects are limited to lactating aggressors).

Furthermore, although knockdown experiments are complicated, take a longer timeframe, and demand more detailed validation (qPCR, in situ hybridization, and so on). Pharmacological manipulations are a rather straightforward approach. Therefore, I strongly recommend that the authors either perform a behavioral pharmacology experiment by injecting a cocktail of PrLR and OTR antagonists into the PMV of aggressor lactating females to link their in vitro (virgin) and in vivo data or restructure their manuscript to shift the focus away from the hormones (maybe shifting this data to the supplementary) and limiting discussing their hormonal effects on cell excitability rather than on behavior.

Thank you for the opportunity to submit a revised version of our manuscript “*Maternal Aggression Driven by the Transient Mobilisation of a Dormant Hormone-Sensitive Circuit*” (NCOMMS-24-19332-T) for consideration for publication in *Nature Communications*. We are grateful to you and to the two Referees for their careful reading of our manuscript and the insightful comments and constructive criticism of our work. In the text below, we discuss these comments and describe how we have responded to them point by point. We hope that the changes that we have introduced are satisfactory and the manuscript now meets the standards for publication in *Nature Communications*.

AUTHORS’ RESPONSE TO REVIEWER COMMENTS

Referee #1

1. *The reviewer appreciates the effort made by the authors to improve the study. The results showing differential excitability of aggressive and non-aggressive lactating females are informative, further supporting a relationship between PMv excitability and aggression. The experiments shown in the paper are done elegantly, and the data quality is high.*

We thank Referee #1 for their positive comments on the experiments, new analysis, and rewriting performed in response to the preceding round of review, and for their kind words about the quality of the work.

2. *However, the manuscript still feels composed of two disconnected parts. Figures 1 and 2 show that PMv excitability is central to maternal aggression, while Figures 3 and 4 show PMv cells can be modulated by oxytocin and prolactin, but these hormones appear to play no role in aggression; instead, they suppress parental behaviors and cause behavioral changes in open field and elevated plus maze. The reviewer believes that it is crucial to tie these two parts together. Specifically, the study will benefit from a functional experiment that demonstrates the role of oxytocin/prolactin in maternal aggression.*

For example, does OXTR antagonist injection into the PMv suppress maternal aggression in lactating moms? Does OXTR or Plr KO at the PMv (a strategy used by the group in previous studies) impair maternal aggression? While the authors showed that oxytocin and prolactin do not increase aggression in virgin females, it remains possible that the hormones will boost aggression in lactation moms. If such a link really could not be found, the authors may consider splitting Figures 1&2 and 3&4 into two papers since they do not support one coherent conclusion.

We understand, and share, the Referee’s interest in the role of OT and Prl in maternal aggression. Without doubt, this is a topic that remains insufficiently understood, and additional work is required for it to

be resolved. The purpose of the experiments in the study that involve these hormones was, however, to connect our findings with the endocrine agents that envelop the maternal state, rather than an exhaustive study of these hormones' precise role and pharmacology in this particular behaviour. Admittedly, there is much more to be done. Our work is a first, not a last, step in defining Prl/OT's interrelationship with the PMv aggression circuit. Yet, we believe that our contribution is significant. In this manuscript we provide evidence that the hormones can act on the relevant "aggression" cells, by which cellular mechanisms, and we propose a framework for their role in triggering/facilitating the expression of the behaviour. We also provide data indicating that a rise in ambient PMv OT/Prl is not sufficient to trigger aggression in the non-puerperal state, but that additional (and as-yet unaccounted) factors need to be mobilized. All of this is highly novel and as we understand it the Referees agree with this assessment.

Our concern regarding the experiments proposed by the Referee, in addition to being beyond the scope of the present study, is not that they demand substantial labour and resources. We consider that the proposed experiments of pharmacological and genetic manipulation of OT/Prl signaling cannot give a satisfying answer to the question being raised about the relative necessity of these hormones (and their actions in the PMv) in dams' attacks. As outlined in our response in the preceding round of revision, this is due to the potential complementary role of the two hormones, compounded by the complex signaling repertoire (multiple receptor subtypes) through which OT can act, which introduces significant combinatorial difficulties. Performing highly complex experiments involving numerous animal procedures (*i.e.*, dams going through multiple pregnancies, stereotactic surgeries, and behavioural paradigms, including exposure to aggression) with limited scientific gain is not something we believe can be justified scientifically or ethically.

The Referee proposes an option to split the ms. in two, and leave out the parts exploring the actions of Prl/OT (presented in Fig. 3). (We should point out, for the sake of correctness, that the Referee states that half of the figures deal with the role of Prl and OT. In fact, it is only Figure 3, reflecting the fact that this is not a dominant message of the study.) However, we see a clear continuity with the other data presented in our manuscript. Firstly, these experiments, while making no claims to be the definitive word on the hormones' role(s) in maternal aggression, play an important role in tying the novel PMv maternal aggression system into the context of these Prl's and OT's well-established roles as framing the maternal state. Secondly, we believe that in our manuscript we present the most in-depth exploration of Prl's effects on neuronal excitability to date - anywhere in the brain, and for the first time reveal the OT system's structural and functional relationship to this key node within the central social network. (A PubMed search for "oxytocin and preammillary" yields only a handful of papers, primarily describing scattered anatomical observations, illustrating the novelty of this work.) - For these reasons, we believe that dividing the manuscript into two would not be meaningful.

We want to assure Referee #1 that we take the concerns they raised seriously, and in response we revised the text to:

a) Render the reader aware that this is a first, not a last, step to connect the PMv aggression system to the hormonal agents that carry motherhood.

b) We stress that additional work is necessary, especially to determine the behavioural correlate of the electrophysiological effects we see, and further contextualize what we think the negative findings applying hormones in virgin female mice tell us about the systems need for additional triggers.

c) We edited the text to emphasize that there could be alternative explanations to account for our findings. We acknowledge that the Referee may still disagree with our rationale, as differences in interpretation are a natural part of scientific discourse. However, we hope they recognize that the Prl/OT data included in our study represent a meaningful step forward, offering key insights that will help guide future research in the field.

3. Lastly, several places state “Data not shown”, please show those data to support the statement.

The Referee requests that we show recordings and quantification where previously stated “data not shown”. We have identified two such instances and have revised accordingly. Thus, we have now included illustrations and quantifications of the T-type Ca^{2+} channel blocker, ML218, on rebound depolarization (Fig. 3g), and the effect of Prl on spontaneous inhibitory postsynaptic currents (Fig. 3j).

Referee #4

1. In the manuscript “NCOMMS-24-19332-T: Maternal Aggression Driven by the Transient Mobilization of a Dormant Hormone-Sensitive Circuit” the authors attempt to dissect the neuronal circuits and the brain hubs regulating maternal aggression in mice. They found out that PMVDAT neurons are activated and seem to be needed during the expression of maternal aggression. Additionally, their ex-vivo data portray a picture suggesting that the hormones oxytocin (OT) and prolactin (PrL), which are known to be upregulated during lactation, are affecting PMVDAT neurons' excitability and may contribute to the PMVDAT regulation of maternal aggression. The manuscript is well-written, the panels and figures are well-displayed and the results seem robust. Although the manuscript has already undergone one round of revisions and the authors provide explanations as well as consistent changes in the text, this reviewer has found some major points which should be addressed before publication.

We thank the Referee for the positive remarks on our work.

2. Introduction, abstract, and general

- In the abstract the authors write “Aggression, a sexually dimorphic behaviour, is prevalent in males and typically absent in virgin females”. That definitely might be the case for C57BL6 strains but it is definitely not the case for other species such as humans, rats, hamsters, other mouse strains, *Peromyscus*, and many others. The authors should be careful to not limit their view of virgin female aggression research to mouse models. Indeed, several studies in the last years have used other species to access virgin female aggression. It comes to my attention that those studies have not been cited by the authors, that feels a bit troubling, especially taking into account that research done in females already gets fewer citations than research performed in males. Among the manuscripts is important to highlight:

Oliveira VEM, Lukas M, Wolf HN, Durante E, Lorenz A, Mayer AL, Bludau A, Bosch OJ, Grinevich V, Egger V, de Jong TR, Neumann ID. Oxytocin and vasopressin within the ventral and dorsal lateral septum modulate aggression in female rats. *Nat Commun.* 2021 May 18;12(1):2900. doi: 10.1038/s41467-021-23064-5. PMID: 34006875; PMCID: PMC8131389.

Oliveira VEM, Bakker J. Neuroendocrine regulation of female aggression. *Front Endocrinol (Lausanne).* 2022 Aug 10;13:957114. doi: 10.3389/fendo.2022.957114. PMID: 36034455; PMCID: PMC9399833.

Newman EL, Covington HE 3rd, Suh J, Bicakci MB, Ressler KJ, DeBold JF, Miczek KA. Fighting Females: Neural and Behavioral Consequences of Social Defeat Stress in Female Mice. *Biol Psychiatry.* 2019 Nov 1;86(9):657-668. doi: 10.1016/j.biopsych.2019.05.005. Epub 2019 May 13. PMID: 31255250; PMCID: PMC6788975.

Terranova JI, Song Z, Larkin TE 2nd, Hardcastle N, Norvelle A, Riaz A, Albers HE. Serotonin and arginine-vasopressin mediate sex differences in the regulation of dominance and aggression by the social brain. *Proc Natl Acad Sci U S A*. 2016 Nov 15;113(46):13233-13238. doi: 10.1073/pnas.1610446113. Epub 2016 Nov 2. PMID: 27807133; PMCID: PMC5135349.

Silva AL, Fry WHD, Sweeney C, Trainor BC. Effects of photoperiod and experience on aggressive behavior in female California mice. *Behav Brain Res* (2010) 208:528–34. doi: 10.1016/j.bbr.2009.12.038

We thank the Referee for their input. It was certainly not our intention to disregard the fact that aggression in females follows species-specific patterns. We have now incorporated the Referee's references and adapted the text in the abstract and in the first paragraph of the introduction to better communicate to the reader that female aggression is indeed exhibited by several species (outlined in the literature provided by the Reviewer).

3. On that same aspect the authors write in the abstract "While maternal hormones are known to elicit nursing, their potential role in maternal aggression remains elusive". That sentence is misleading and undermines the work of researchers such as Prof. Oliver Bosch, Prof. Inga Neumann, and Prof. Benjamin C. Nephew who have worked for the last decades investigating how 1) peptide signaling, 2) peptide release and 3) receptor binding densities affect maternal behaviors including maternal aggression. Although the authors cite some of the reviews written by those researchers, it would be important to give credit to original papers and of course, change this sentence in the abstract and make corrections throughout the manuscript.

While it has certainly been our sincere goal to accurately reflect the state of the literature, some references may have been missed that speak to the association of OT and Prl with maternal aggression. We thank the Referee for pointing out (and agree) that the work by the investigators listed has indeed given vital insight to this issue. Thus, we adapted the sentence from the abstract cited by the Referee to: "In addition, while maternal hormones are known to elicit nursing, their role in maternal aggression, in particular with regard to target sites and cellular mechanisms, remains elusive."

Furthermore, we have included additional references in the Introduction highlighting the work of Drs. Bosch, Neumann and Nephew, to render the reader better aware of their seminal contributions: OJ Bosch ... ID Neumann, 2005. *J Neurosci*;25:6807-15 and BC Nephew *et al.* 2009. *Behav Neurosci*;123:949-57.

4. My third general point is regarding the translational relevance of maternal aggression. As an animal researcher myself, I do understand that scientists have to work with the models they have at hand. Also, every animal model has its gains and pitfalls. Although, maternal aggression is an ecologically relevant model for understanding the motivational aspects underlying aggression in rodents that might not be the case for other

species including humans. Women exhibit aggression outside of parturition/lactation/pregnancy. Additionally, pregnancy and puerperium have not been tied to the prevalence of aggression disorders in humans. Taking this scenario into account this reviewer invites the authors to critically look into the use of maternal aggression to study the neurobiology of aggression and sex differences in general, as the authors themselves wrote, lactation is a unique physiological period for females in which several systems are upregulated. Thus, one should bear in mind that the maternal brain and behavior may not completely reflect the female brain and therefore cannot directly be compared to the male brain and behavior (aggression). Finally, this point does not aim to undermine lactation as a powerful scientific tool but rather highlights how important this model is to understand the maternal brain, which does not necessarily reflect the non-lactating female brain which is the target of sex differences research. The reviewer would like to know the thoughts of the authors on the topic and have that addressed in the discussion.

We could not agree more with the Referee. Indeed, aggression is a complex behaviour that is triggered by different stimuli and takes different expressions not only between species and between males and females of the same species, but also between members of the same sex. Such is the case of the surprisingly stable phenotype of male aggressors vs. non-aggressors in the resident-intruder paradigm, and – in the case of maternal aggression – within an individual under different life circumstances. Indeed, the questions that motivated us to perform this study were not only a desire to understand the neuronal underpinnings of maternal aggression in mice, but rather, how a behaviour can be accessible to an individual under one (brief) part of its life, and not of that individual's regular behavioural repertoire. In addition, we were interested in what type of brain mechanisms might underlie this remarkable transition. Maternal aggression in female mice offers an ideal platform to address this conundrum. We also emphasize that the conclusions of this study speak to the idea (as *e.g.* championed in Catherine Dulac's seminal work) that there may be a similar blueprint for behaviours in male and female brains, such that a behaviour can be called upon when ethologically useful, rather than having different hard-wiring for each sex. Under this theoretical framework, the maternal brain is indeed not a model for the non-lactating female brain, but serves to illustrate the remarkable plasticity underlying adaptability in social behaviours and the circuit modulation that may be involved in this.

We have expanded our discussion to address these nuances. We now emphasize that maternal aggression is a powerful model to uncover the neural circuitry and hormonal mechanisms specific to the maternal state, and that it does not reflect the full spectrum of aggression exhibited by females in other contexts. Rather, it provides a window into how adaptive, state-dependent modulation of aggression can occur, potentially offering insights into shared cellular and circuit-level processes that might be relevant across different contexts of aggression (please see changes in abstract and introduction text).

5. In the introduction the authors write “Importantly, it is not well understood if the neural modules that lead to the expression of aggression are shared between the two sexes, or exhibit sexual dimorphism.” In fact, there are not so many studies in females and on sex differences, that is another reason why the authors should cite studies such as Terranova 2016, Newmann, 2020, and Oliveira, 2021 which actually address this question.

The references listed by the Referee have been added to the Introduction to highlight earlier work on sex differences.

6. Results, methods, discussion

- Control experiments in virgin females are normally performed in estrus females. What is the rationale for that? Why was this specific phase of the estrous cycle chosen? Especially taking into account that prolactin is released during the evening of proestrus which antecedes estrus.

The Referee raises the important questions regarding our choice of control animals. Admittedly, “control” is in some ways an imperfect term – in a sense we are using these animals more for “comparison”, as there is not a single type of animal that would function as a universally accepted “control” for the state of pregnancy. However, to follow standard practice and not confuse semantics, we have kept the term “control” in the ms. That said, in our study, we elected to use virgin females in estrus for comparison as this phase is characterized by relatively high circulating prolactin levels. Although prolactin release indeed peaks during proestrus, levels remain elevated into estrus (as shown in our quantification [Fig. 3a]). Since lactating dams exhibit high prolactin levels which we suggest as critical for maternal behaviors – including maternal aggression, using estrus-phase females provides a closer physiological comparison. This choice lessens the difference in baseline prolactin levels between our control (virgin) group and lactating dams.

7. Additionally, the work of Alexa Veenema shows that females in estrus exhibit changes in OXTR densities in hypothalamic regions. Could those factors have influenced the behavioral and ex vivo results of the authors? Why not use diestrus females?

We agree with the Referee about the importance of the findings by Veenema and colleagues on estrous phase-related changes in OT-binding. However, to our knowledge those studies did not identify (or focus on) the PMv as a locus for receptor modulation. Thus, it is difficult to speculate if changes in receptor activation over the cycle may have influenced our results, though this possibility can, of course, not be fully excluded.

8. Finally, regarding the behavioral perspective, Oliveira 2021 has found that optogenetic stimulation of OXT neurons in the PVN and SON as well as OXT terminals in the lateral septum of virgin female rats failed to elicit aggression when females were either in proestrus or estrus but triggered aggression when females were in metestrus or diestrus. Moreover, estrus has been shown to decrease aggression in virgin females in a couple

of studies (to a variable extent). Could direct optogenetic stimulation trigger aggression in diestrus females in the present study? Could the lack of effect reflect an estrous cycle interaction?

The work by Oliveira *et al.* performed in rats is admittedly of great interest. However, as stated above, for the current study, our rationale was to use animals where prolactin was at its highest (non-pregnant) level (*i.e.*, the proestrus/estrus phase). This point, and the two points preceding it, do, however, highlight that the selection of a comparison estrous cycle state for pregnancy has no unambiguous “right answer”. (Of course, a full evaluation of all cycle stages would shed light on cycle variation, but is not practically possible for a study of this scope.) We thank the Referee for bringing this important issue up for discussion, and we added a sentence to the Discussion to bring the reader’s attention to this aspect.

9. On the same topic, this reviewer noticed that the authors have used juvenile females as intruders for the loss-of-function experiments for lactating females (maternal aggression) in order to enable optimal (high) aggression levels but adult intruders for virgin females. Why did the authors choose this approach? Why not use juveniles for both protocols? Especially knowing that Hashikawa, 2017 and Oliveira, 2019 have shown that virgin female mice and rats, respectively, preferentially attack juveniles. What would be the outcome of performing aggression training using juvenile females as intruders? Would that lead to aggression and activation of PMVDAT neurons?

For these optogenetic experiments, the design was chosen (agnostic to outcome) relative to *manipulation* rather than *subject identity*; *i.e.* for all gain-of-function experiments, irrespective of if the resident was a pregnant dam or a virgin female, we used adult intruders. This design was selected to ensure that we could compare results yielded under identical test conditions. For the loss-of-function experiments (only performed on dams) we used juvenile intruders to make certain that we did not miss any aggression that may have been there; we deemed this design necessary to be able to evaluate the necessity of the PMv^{DAT} neurons for the expression of maternal aggression. It cannot be excluded that PMv^{DAT} stimulation (gain-of-function) of virgin females in a scenario where they meet a “weaker” intruder (*i.e.* juvenile) would have triggered expressions of aggression, but this would not have allowed for a direct comparison with dams, which was the purpose of that experiment.

To better contextualize these data, we have revised the Discussion to add the caveat “(It cannot be fully excluded that there may be circumstances when virgin female mice could mount an attack towards a patently weaker conspecific in response to exogenous PMvDAT stimulation, as we did not apply optogenetic stimulation in an encounter towards juvenile intruders, which more effectively trigger aggression in females^{9,12}. Even in that scenario, however, the conclusion would remain that the bar to elicit attack has been at a minimum significantly lowered, if not altogether removed, with the advent of pregnancy)”.

10. Regarding behavioral outcome how do loss- and gain-of-function experiments affect other behavioral domains such as non-aggressive social investigation and home cage exploration? This data might be relevant, especially knowing that activation of PVMDAT neurons does not seem to be aggression- but rather social investigation-specific.

We appreciate the Referee's question regarding the broader behavioral impact of manipulating PMv^{DAT} activity in dams. To address this issue, we assessed how these neurons influence non-aggressive social investigation and exploration using loss-of-function experiments (Revised Fig. S2c-d, and statement added in Results).

In the three-chamber sociability test, we presented dams with two non-aggressive social investigation options, specifically: 1) the dam's own pups, and 2) a young female conspecific. Here we found that genetic deletion (through Casp3 expression) of PMv^{DAT} cells dysregulates the native animal social behavioral repertoire. Specifically, it switches the animal's behavior from attending to the pups, to spending more time with a novel female conspecific. These results, although nuanced, support the idea that PMv^{DAT} neurons are involved in the prioritization of behavior, a concept we explored in multiple points across the manuscript (*e.g.*, Fig. 4 and Fig. S3).

Genetic ablation of PMv^{DAT} neurons led to no changes in the open field test and exploration related variables including distance moved during the assay, and time spent close to the border of the arena.

11. Going back to the previous work of the authors on winning experience, the authors have shown that testosterone could generate the needed plasticity to induce winning in males. In females, it does not seem to be so simple as treatment with OT and prolactin chronically failed to affect aggression. However, **would direct stimulation of PMV DAT of non-aggressive lactating mice turn those females into aggressors?** In the same regard, could direct stimulation of PMV DAT neurons lead to maternal aggression even after OT and prolactin receptor antagonist treatments? These experiments could provide evidence as to whether PMV DAT neurons are downstream OT and Prl. Also were there differences in prolactin and OT levels between aggressors and non-aggressors lactating females?

We should first point out that while we have previously published a study on the role of PMv^{DAT} cells in the competition for social hierarchies in males (Stagkourakis et al., 2018), we have not, as implied by the Referee, addressed the role of testosterone in this context. But, we agree that it is important to determine if direct stimulation of these neurons in non-aggressive dams can elicit attacks. Prompted by the Referee's suggestion, we have performed this experiment, and the results are shown in Fig. S1h, I (and statement added in Results and in Discussion). Importantly, optogenetic activation of PMv^{DAT} neurons in inherently non-aggressive lactating dams did not lead to attacks or other expressions of aggression. This finding resonates with our observations and those of others in recent years, that an animal does not show aggression naturally

(as is the case in virgin female estrus mice and non-aggressive lactating dams [the present paper], as well as non-aggressive male mice [Golden *et al.*, 2016, Stagkourakis *et al.*, 2018 and Stagkourakis *et al.*, 2020]), optogenetic stimulation of an aggression-related neuron population is not sufficient to trigger its expression. This aspect supports the proposal that additional, permissive factors, are required for the full transformation into an aggressor phenotype, as commented upon in the Discussion. – The Referee also suggests other experiments to further define the exact role of Prl and OT. We agree that these are important questions to answer in future studies, but lie outside the scope of the present study, as discussed in our response to Referee # 1, Item 2.

12. Concerning the ex vivo patch-clamp data, I know that the authors were asked by the other reviewers about what would be the effect of treating lactating females with prolactin and oxytocin, to what they answered: “The recordings were performed on slices from virgin female mice, as our hypothesis is not that PMvDAT cells become sensitive to Prl and OT only after giving birth, but rather that the dramatic increase in circulating levels of these hormones makes their action all the more potent. The reason for not using dams for these recordings was the risk that sustained exposure to high concentrations of Prl and OT may have rendered them insensitive to the exogenous applications of these hormones, which would have confounded the results. Adaptation is, in our experience, common in electrophysiological experiments using peptides/polypeptide ligands, and the high degree of STAT5 phosphorylation in PMvDAT neurons in lactating dams (present data) strongly indicates that they are under substantial Prl receptor activation. It was, however, not clear in the original submission that these experiments were performed on tissue from virgin female mice. We apologize for this missing information and have now specified it in the Results.” I disagree with the author’s point indeed lactating females show increased levels of prolactin and oxytocin but only up to stimulation (suckling). That does not mean those neurons will be hypersensitive to OT or Prl and that the levels of the hormones will be high in the brain during the whole time. Can the authors actually provide literature references for that? Additionally, the authors missed the opportunity to generate the missing link between peptide signaling and excitability changes by treating lactating female slices with OT receptor and prolactin receptor antagonists alone and in combination to investigate whether this could reverse the plasticity changes seen in the lactating brain.

We thank the reviewer for raising these issues and the opportunity to clarify our experimental approach regarding the *ex vivo* patch-clamp recordings. In our study, we deliberately performed recordings on slices from virgin female mice rather than lactating dams.

Although prolactin release in lactating females is stimulated during suckling (Bridges *et al.*, 1985), the basal levels of prolactin remain elevated even in thelectomized female rats following birth of their pups (Moltz *et al.*, 1969). Importantly, sustained or repeated exposure to high concentrations of prolactin *in vivo* (as evidenced by the high degree of STAT5 phosphorylation in PMv^{DAT} neurons in lactating dams; Brown *et al.*,

2010) can lead to receptor desensitization or adaptation. Such adaptation is documented in electrophysiological studies using peptide or polypeptide ligands (e.g., Augustine *et al.*, 2017; Brown *et al.*, 2012).

Our concern was that slices from lactating dams, which have already experienced sustained high hormone levels, might display altered sensitivity to additional exogenous application of prolactin and oxytocin. By using virgin female slices—where baseline exposure to these peptides is minimal—we aimed to unambiguously assess the acute effects of these hormones on PMV^{DAT} neuron excitability without the confound of prior receptor adaptation.

We have now added text in the Results section rendering the reader aware of our rationale and included relevant literature, since we agree with the Referee that this experimental choice merits elaboration. – The Referee also proposes an experiment where receptor antagonists are applied to *ex vivo* slices to determine the role of the ligands (Prl/OT) on plasticity. The rationale for such an experiment is not fully clear to us for two reasons. Firstly, we do not expect significant levels of the cognate hormones to be present in the slices (and thus no receptor action that could be blocked by antagonists). Secondly, in the model we propose based on our findings, we are not hypothesizing plasticity (remodeling) actions by Prl and/or OT, but rather acute actions superimposed on plasticity driven by other mediators than these hormones. Nevertheless, we do agree, as expressed above, that further research will be needed in the future to determine the exact role of Prl/OT in maternal aggression.

13. Also regarding ephys data the authors find an increase iPSCs after prolactin treatment, how do the authors think this data adds to the idea of prolactin turning PMVDAT neurons more excitable?

We understand the Referee's point that an increase in synaptic inhibition may seem counterintuitive given the argument that Prl excites PMV^{DAT} neurons. Certainly, the net effect of the hormone on these cells is to depolarize them and drive towards spiking, as shown in Fig. 3e. All PMV^{DAT} cells recorded through *ex vivo* patch-clamp electrophysiology were depolarized in a reversible manner (Fig. 3e). The increase in excitatory inputs (sEPSCs) likely contributes to the depolarization effect, along with the direct, postsynaptic actions that we describe. Yet, there is also a concurrent augmentation of inhibitory impulse traffic when Prl is applied (but not sufficient to result in a net inhibition). To our understanding, this is not unexpected, as homeostatic mechanisms often counterbalance heightened excitability by engaging local inhibitory circuits, something that has been observed widely in the brain, including in neighboring hypothalamic circuits (Tao *et al.*, 2024; Zhu *et al.*, 2024). Future work could further dissect how prolactin shifts the excitation-inhibition balance at the network level. To render the reader aware of this nuance in the Prl's electrophysiological effects, we added text in the Discussion ("*Notably, while the net effect of prolactin is excitatory, it increases both excitatory and inhibitory synaptic inputs onto PMV^{DAT} neurons, which could reflect a homeostatic mechanism recruited to*

balance the elicited heightened excitability.”) – As requested by Referee # 1, a quantification of the Prl-induced increase in sIPSCs has now been integrated in Fig. 3j.

14. Crosstalk between OT and vasopressin receptors has been shown including in the context of female aggression (Oliveira 2021, Tan 2020). To the comment of the other reviewers, the authors claim “Thus, although available histochemical data indicate the expression of OT-R’s, but not vasopressin receptors in the PMv these pharmacological results suggest that vasopressin receptors may also contribute to the peptide’s action on PMvDAT neurons”. Here the authors use a rat reference but they should keep in mind that the patterns of OT and VP receptor expression differ between mice and rats. Meaning that mice could express receptors in different locations and densities than rats. On that aspect, using a V1aR antagonist plus OT treatment could help to elucidate the non-OT receptor-mediated mechanisms described in the manuscript. Additionally, vasopressin itself has been shown to also modulate maternal aggression in rats, could the authors speculate whether vasopressin acts synergically with OT and Prl in their model?

We fully agree with the Referee that caution is warranted when comparing mRNA expression between species. Regrettably (and to our surprise), we were unable to find extensive data on the anatomical expression of OT and VP receptor expression in the mouse, and thus cited findings also from rat studies as a proxy, albeit an imperfect one. This caveat is now emphasized in the Results. It should be noted that *in situ* hybridization for the *Avpr1b* receptor, as documented by the Allen Institute (<https://mouse.brain-map.org/gene/show/26109>), also supports the idea that vasopressin receptors are sparsely found in the peri-PMv region with low to zero levels in PMv. In the Discussion, however, we have added a sentence stating that the oxytocin effects we observe may arise, at least in part, through vasopressin receptors, so that the reader is aware of this possibility: “*A similar increase in excitability is induced by OT (possibly acting, at least in part, via vasopressin receptors), indicating a synergistic action of the two hormones.*” Given that there exists an intriguing literature connecting vasopressin to aggression, this is an issue that merits future exploration. We thank the reviewer for their input.

REVIEWER COMMENTS

Reviewer #1 (Remarks to the Author):

The reviewer wished that the authors could have addressed the PMv OXTR and PRLR role in maternal aggression more thoroughly but recognized that this requires substantial more effort. While a lot more could be done to fill the gaps in the paper, the reviewer will not request more experiments that will delay the publication of the study. The reviewer hopes that the authors will continue their effort to further elucidate the functional importance of PMv OXT and PRL signaling in maternal aggression.

We thank the Reviewer for their assessment that no further experimental work should be required for publication. The Reviewer's concerns are noted, and we acknowledge that the current study raises important new questions that will require additional investigations before the final verdict on the role of the PMv PRLR and OXTR in maternal aggression can be passed. We take these comments to heart as we plan the next steps of our research in this field.

Reviewer #4 (Remarks to the Author):

First of all, I would like to thank the authors for i) carefully answering all my points, ii) performing one extra experiment in the non-aggressor lactating females, which does show that PMVDAT neuron excitation is not sufficient to trigger maternal aggression but rather that some contextual changes, presumably high levels of OXT and Prl, are needed in combination to stimulate maternal aggression and iii) for assessing the role PMVDAT in social preferences. I acknowledge that this is an extensive amount of extra work and that many points are now clearer. However, I still have one major concern regarding the link between hormones, PMVDAT neurons, and maternal aggression.

We thank the Reviewer for their acknowledgement of the amount of work that went into the latest revision of our study.

I did read the answer of authors to Referee #1, and I can completely relate to the fact that performing extra experiments to connect OT and Prl to the PMVDAT neurons can be extremely time-consuming and might even raise further questions.

Additionally, I also understand the point the authors raised regarding my question about treating lactating female slices with PrL and OT.

We appreciate the Reviewer's understanding of the complications inherent to the experiments proposed in the last round of revisions.

However, the title of the manuscript itself as well as the discussion imply a pivotal role of both hormones in the effects reported by the authors in maternal aggression. Please see below:

"Maternal Aggression Driven by the Transient Mobilisation of a Dormant Hormone-Sensitive Circuit"

Respectfully, we believe that the title is indeed consistent with the results presented in the manuscript. Specifically, with regard to Prl and OT, no other claim is made than that the PMv^{DAT} neurons are sensitive to these hormones, which is fully supported by the electrophysiology and the pSTAT5 data. The title does not imply a causal role for Prl and/or OT in maternal aggression mediated by PMv^{DAT} neurons.

“With parturition, the surge in Prl and OT provide a powerful depolarising influence synergising with this primed state to push the circuit into hyperactivity and thus drive the full expression of aggression. Such a role for Prl and OT in maternal aggression would also account for the disappearance of this behaviour with weaning, when serum levels of these hormones quickly drop to pre-puerperal levels”

“Our results suggest a model in which PMvDAT neurons are hormonally primed into a hyperexcitable state in the lactating dam, greatly lowering the threshold for eliciting attacks against conspecifics.”

“The hallmark hormonal agents of lactation, Prl and OT, can powerfully, and through multiple convergent mechanisms, stimulate these cells into discharge mode. These findings suggest a flexible neural framework for hormonally driven generation of adaptive aggression”

Again, respectfully, the three examples above are all clearly prefaced to the reader as a proposed model. While one can have different views on how far the boundaries of speculation are stretched, in the present manuscript we present a model consistent (albeit not yet fully validated) with the data and the literature. But a Discussion that does not make an effort to contextualize the findings is of little service to the reader (the reader will, of course, have to make their own judgment to what extent they find the model reasonable). In our original manuscript and in subsequent revisions, aided by the insightful comments of the four Reviewers, we have made our best effort to point out to the reader that additional work is needed to fully understand the role of the PMv^{DAT} neurons in general, and their sensitivity to Prl and OT in particular, in murine maternal aggression.

Although the authors claim in their answer to Referee 1# “The purpose of the experiments in the study that involve these hormones was, however, to connect our findings with the endocrine agents that envelop the maternal state, rather than an exhaustive study of these hormones” they do use both hormones to speculate on their findings without providing data that sustain those statements. This is misleading, taking into account that those experiments were performed in different groups (hormonal in vitro findings as well as the chronic peptide treatments in virgins, whereas behavioral effects are limited to lactating aggressors).

Furthermore, although knockdown experiments are complicated, take a longer timeframe, and demand more detailed validation (qPCR, in situ hybridization, and so on). Pharmacological manipulations are a rather straightforward approach. Therefore, I strongly recommend that the authors either perform a behavioral pharmacology experiment by injecting a cocktail of PrLR and OTR antagonists into the PMV of aggressor lactating females to link their in vitro (virgin) and in vivo data or restructure their manuscript to shift the focus away from the hormones (maybe shifting this data to the supplementary) and limiting discussing their hormonal effects on cell excitability rather than on behavior.

We thank the Reviewer for their interest in the role of Prl and OT in maternal aggression, which we fully share. As detailed in our response to the previous review, we have significant concerns about the ability of currently available pharmacological tools to conclusively identify a causal role for OT and/or Prl within the PMv in maternal aggression. We do understand the Reviewer’s concerns about this issue, however, and agree that a restructuring of the Discussion can help the reader more fully comprehend that the role of these hormones in this nucleus can only be fully understood with additional future work, and that while our experiments pertinent to this issue primarily (albeit not exclusively) involve observations from *ex vivo* recordings from non-lactating female mice, the link to behaviour in dams has not yet been established experimentally. We have rewritten the closing paragraphs of the Discussion to stress this aspect.